# WINDOW ATTENTION IS BUGGED: HOW NOT TO INTERPOLATE POSITION EMBEDDINGS

**Daniel Bolya**[1,2*]     **Chaitanya Ryali**[2]     **Judy Hoffman**[1]     **Christoph Feichtenhofer**[2]

[1] Georgia Tech      [2] FAIR, Meta

{dbolya,judy}@gatech.edu, {chayryali,feichtenhofer}@meta.com

## ABSTRACT

Window attention, position embeddings, and high resolution finetuning are core concepts in the modern transformer era of computer vision. However, we find that naïvely combining these near ubiquitous components can have a detrimental effect on performance. The issue is simple: interpolating position embeddings while using window attention is *wrong*. We study two state-of-the-art methods that have these three components, namely Hiera and ViTDet, and find that both do indeed suffer from this bug. To fix it, we introduce a simple absolute window position embedding strategy, which solves the bug outright in Hiera and allows us to increase both speed and performance of the model in ViTDet. We finally combine the two to obtain HieraDet, which achieves 61.7 box mAP on COCO, making it state-of-the-art for models that only use ImageNet-1k pretraining. This all stems from what is essentially a 3 line bug fix, which we name "absolute win".

## 1 INTRODUCTION

Transformer-based architectures dominate many tasks throughout computer vision (Zhai et al., 2021; Li et al., 2022a; Kirillov et al., 2023). But despite their ubiquity, these architectures are relatively new, and thus many best practices have yet to be set in stone. In this paper, we focus on a relatively unassuming operation in modern vision transformers: *interpolating position embeddings*.

Absolute position embeddings added at the beginning of a transformer allow the model to distinguish between tokens based on location, an important detail for most vision tasks. While natural language processing (NLP) can sometimes get away with omitting a position embedding (Haviv et al., 2022), those in computer vision often find themselves adding *additional* position embeddings to deal with more difficult tasks (e.g., detection in Li et al. (2022a)). Several recent works even design custom position embeddings to inject directly into attention (Liu et al., 2021b; Graham et al., 2021; Li et al., 2022b), so that location information is right where it is needed.

However, these additional position embeddings can be costly: techniques like attention bias (Liu et al., 2021b; Graham et al., 2021) or relative position embeddings (Li et al., 2022b;a) are added directly into the attention matrix. Not only are these operations slow, but they also cannot benefit from recent innovations such as Flash Attention (Dao et al., 2022; Dao, 2023) that speed up transformers by not constructing the attention matrix. Ideally, we would like to avoid relative position embeddings and just use simple and fast absolute position embeddings like the original ViT (Dosovitskiy et al., 2020).

So, why do most architectures *not* use absolute positioning embeddings? One potential reason becomes apparent when we study Hiera (Ryali et al., 2023), a modern hierarchical vision transformer that only uses absolute position embeddings. Hiera is as powerful and more efficient than other state-of-the-art vision architectures, while being composed entirely of simple ViT blocks. Instead of adding more position bias architecturally (e.g., with relative position embeddings), it *learns* good spatial biases through a strong pretext task (i.e., MAE (He et al., 2022)). This makes it the perfect case study for modern architectural design paired with simple absolute position embeddings.

And immediately, an issue presents itself: *Hiera does not **interpolate** well*. When finetuning Hiera on images that are even slightly larger than what it was trained on, the accuracy of the resulting model plummets. This includes mediocre results on detection, where ViT outperforms Hiera (Ryali et al., 2023). The culprit, we discover, is the interaction between *window attention* and *absolute position embeddings*. That is, having window attention and absolute position embeddings in the same model

---

[*]Work done during an internship at Meta. Code and models at https://github.com/facebookresearch/hiera.

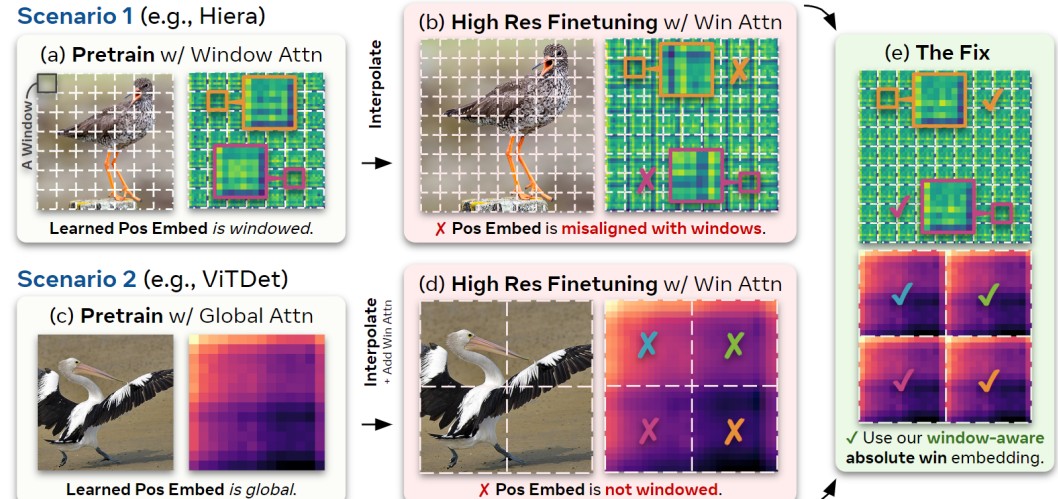

Figure 1: **Window Attention is Bugged** when used with absolute position embeddings and finetuned at high resolution. We study two methods that suffer from this bug: Hiera (Ryali et al., 2023) and ViTDet (Li et al., 2022a). For each, we show a channel of their position embeddings. (a) Hiera's embedding learns repeated patterns aligned with window attention. (b) Interpolating breaks alignment. (c) ViTDet pretrains with global attention, adding windows for finetuning. (d) After interpolating, each window only has a piece of the full position embedding, despite each using the full attention operation from pretraining. (e) We introduce **absolute win**dow embeddings (Fig. 3) to fix both.

induces a **bug** when interpolating to larger images. To solve this, we introduce a simple absolute window-aware position embedding (namely, "absolute win") that can be interpolated to any image size without issue. This change alone is enough to completely alleviate any problems with image size in Hiera, resulting in strong performance on both image and video benchmarks at high resolution.

However, the problem goes deeper: this is an issue in *any* model that uses both absolute position embeddings and window attention at once, e.g., ViTDet (Li et al., 2022a), a state-of-the-art method for detection without extra detection data. In fact, Li et al. find that adding relative position embeddings is necessary for good performance, and we believe this bug is the reason. By using our absolute win method for detection, we can remove almost all the relative position embeddings in the model. This allows for an up to 43% speed-up over the original model while *increasing mAP*. Furthermore, fixing the bug unlocks other techniques that allow us to increase Hiera's performance on detection by up to **1.5 mAP** on COCO (Lin et al., 2014), significantly outperforming ViT and establishing a new state-of-the-art for detection and instance segmentation with only ImageNet-1k pretraining.

Note that we do not claim to introduce any exceedingly novel techniques here. Instead, we identify and analyze a bug present in the current state-of-the-art, introduce a simple strategy to fix it, and establish best practices for interpolating position embeddings. We then show how this alone is enough to improve the state-of-the-art in image and video classification, detection, and instance segmentation.

## 2 BACKGROUND AND RELATED WORK

In this paper, we explore a bug caused by the combined use of three popular components in computer vision: window attention, absolute position embeddings, and high resolution finetuning.

**Window Attention.** In transformers, global attention (Vaswani et al., 2017) compares each token to *all* other tokens. This is the default for ViTs (Dosovitskiy et al., 2020), but it is often costly and potentially wasteful for larger image sizes. Several alternative attention approaches have been proposed for vision, such as pooling attention (Wang et al., 2021; 2022; Fan et al., 2021; Li et al., 2022b) or linear attention (Choromanski et al., 2020; Wang et al., 2020; Bolya et al., 2022). However, particularly popular is window attention (Liu et al., 2021b; Chu et al., 2021; Dong et al., 2022; Li et al., 2022a; Ryali et al., 2023), which compares each token *only* to those in the same window as itself. This makes compute a function of window size, rather than image size, making it much faster.

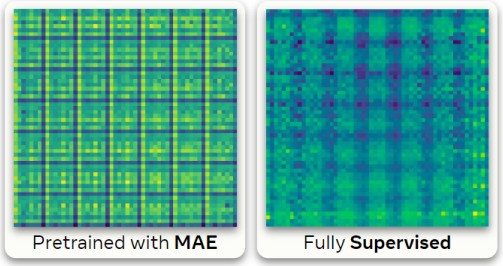

Figure 2: **Abs Pos Embed for Window Attn.** A channel from the *learned* absolute position embeddings in Hiera (Ryali et al., 2023), a simple hierarchical ViT that uses window attention.

| model | pos embed | in-1k acc @ ft res | |
| | | 224px | 256px |
| --- | --- | --- | --- |
| Hiera-L$_{MAE}$ | absolute | 85.6 | 85.2 (- 0.4) |
| | **absolute win** | **85.7** | **86.0** (+0.3) |
| Hiera-B$_{Sup}$ | absolute | 81.8 | 81.4 (- 0.4) |
| | absolute win | 82.4 | 82.7 (+0.3) |

Table 1: **Finetuning at higher res** than the default (224px) fails when using a Hiera model with its default absolute pos emb. Absolute win fixes this for MAE trained models and increases performance for fully supervised (Sup) models.

**Absolute Position Embeddings.** By default, attention has no spatial bias, i.e., each token is treated the same regardless of its position. The original transformer (Vaswani et al., 2017) as well as ViT (Dosovitskiy et al., 2020) introduce spatial bias using an *absolute position embedding* (abs pos embed) different for each spatial location and added directly to each token after the embedding or patchification step. Subsequent strategies have been proposed that add this bias directly to the attention layers, such as relative position embeddings (Shaw et al., 2018; Huang et al., 2020; Liu et al., 2021a; Wu et al., 2021), attention bias (Liu et al., 2021b; Graham et al., 2021), or rotary embeddings (Su et al., 2021). While these are effective, they are often much slower than simply adding an absolute embedding at the start of the network. Furthermore, works such as Hiera (Ryali et al., 2023) show that relative position embeddings (relpos) are not necessary so long as the model is trained effectively.

**High Resolution Finetuning.** Finetuning models at higher resolutions (high res) has been a common practice in computer vision in the past (Huang et al., 2019; Tan & Le, 2019), so it's not a surprise that the same is true for modern vision transformers (Liu et al. (2021b); Li et al. (2022b), etc.). In fact, the only component that depends on image size in most ViTs is the position embedding. The standard strategy to resize, e.g., an absolute position embedding is to simply upsample them with bicubic interpolation as in Dosovitskiy et al. (2020). However, in this paper we show that doing this in an architecture that has both window attention and absolute position embeddings can lead to errors.

## 3 DISCOVERING A BUG

We begin our exploration of absolute position embeddings with Hiera (Ryali et al., 2023), a modern hierarchical (i.e., *multi-scale*) vision transformer that forgoes architectural bells-and-whistles in favor of learning spatial bias through a strong pretext task (MAE, He et al. (2022)). This includes using absolute position embeddings like ViT (Dosovitskiy et al., 2020) instead of more complex relative position embeddings (Li et al., 2022a;b) or attention bias (Liu et al., 2021b; Graham et al., 2021).

While Hiera works well in its default configuration, we obtain a puzzling result when finetuning the model on even a slightly larger image size. If we take an original Hiera-L model pretrained on 224px images and finetune it on 256px images for ImageNet-1k (Deng et al., 2009), the top-1 accuracy *drops* by 0.4% (see "absolute" in Tab. 1). Needless to say, this is the *opposite* of what's supposed to happen: prior work significantly benefits from finetuning on larger images.

**The problem.** For efficiency, Hiera uses "mask unit" attention within the first two stages of the network. This is essentially window attention, but each window always corresponds to the same $8 \times 8$ tokens in the input (i.e., the windows get smaller as the tokens are pooled). This results in some interesting emergent properties when used alongside learned absolute position embeddings.

In Fig. 2, we show two channels from Hiera's learned position embeddings, one from the public Hiera-B model pretrained with MAE, and the other from training Hiera-B fully supervised from scratch. In both cases, the learned position embeddings are *tiled*. That is, there is a repeating $8 \times 8$ pattern in the embedding. And unsurprisingly, this has the same layout as Hiera's window attention.

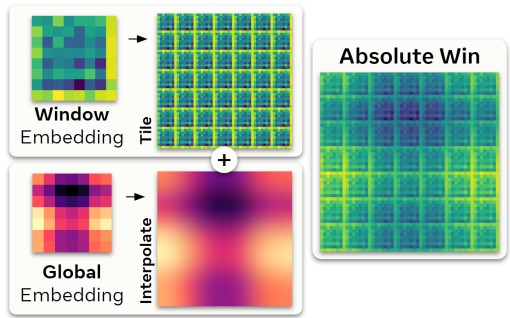

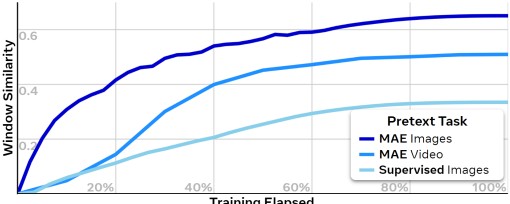

Figure 3: **Absolute Win** embraces the behavior in Fig. 2 by explicitly learning position embeddings as the sum of a tiled *window embedding* and an interpolated *global embedding*. See Appendix D for more learned embedding examples.

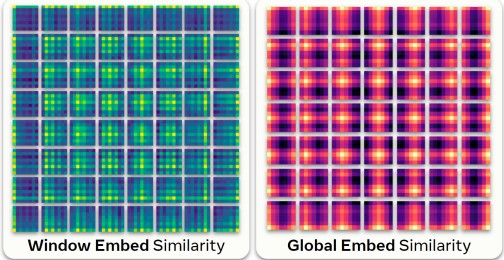

Figure 5: **Absolute Win Similarity.** Cosine similarity between tokens in the learned window and global embeddings (Fig. 3). E.g., the top left patch for window embed compares the top left token in the window embedding with all other tokens in the window embedding.

Figure 4: **Window Repetition** increases over training for the baseline Hiera models. Here we measure the average pairwise cosine similarity between the position embedding of each window.

| position embedding | | in-1k acc @ ft res | |
|---|---|---|---|
| method | global size | 224px | 256px |
| **absolute win** | $14 \times 14$ | **85.7** | 85.9 (+0.2) |
| | $7 \times 7$ | **85.7** | **86.0** (+0.3) |
| | $3 \times 3$ | 85.5 | 85.9 (+0.4) |
| global only | $7 \times 7$ | 85.5 | 85.8 (+0.3) |
| window only | none | 84.8 | 85.1 (+0.3) |

Table 2: **Varying Global Size.** We default to a global embedding of $7 \times 7$ for the global embedding in Tab. 1. This turns out to be optimal for both 224px and 256px acc (Hiera-L$_{\text{MAE}}$).

Normally, this would not be an issue. However, when finetuning on larger image sizes, it is standard practice in vision to *interpolate* the position embeddings (i.e., via bicubic interpolation). But in this case, naïvely interpolating *misaligns* the learned embeddings for each window with the windows themselves (see Fig. 1ab). In most cases, this drastically changes the position embeddings for each window, which explains the low accuracy after finetuning in Tab. 1.

**Why it happens.** Position embeddings add *spatial bias* to the model, which allows attention to learn operations that depend on a token's position. Thus, learned position embeddings are closely tied to the behavior of attention. Since each window shares the same weight matrix (for $Q$, $K$, and $V$), any update made to attention in one window would affect all other windows as well. This would cause the behavior of attention to *average out* for each window during training. And if position embeddings are closely tied to this behavior, then it is not surprising that they would become similar as well.

However, that is not the full picture: the tiling effect is *significantly* more pronounced in an MAE trained model than a fully supervised one (Fig. 2). MAE *deletes* most windows during pretraining, which seems to encourage the model to learn *redundant* operations for each window. That is, each window should be treated the same no matter the position, as not all windows are guaranteed to exist in the forward pass. This produces a starkly different embedding, which suggest that MAE pretrained and fully supervised models are learning *completely different* attention operations.

To motivate this hypothesis, in Fig. 4 we plot the average pairwise cosine similarity of the windows in Hiera-L's position embedding throughout training. For both images and video, pretraining with MAE causes the window similarity to increase drastically over the course of training (though not to 1.0, because Hiera still has global attention as well). Video reaches lower similarity, perhaps because video compression makes discerning the precise location of a token not as beneficial. In contrast, a fully supervised image model *does* increase in similarity, but not as much. The trend suggests each model relies on spatial information differently, with supervised models potentially not enough.

**The fix.** While not necessarily a problem itself, Hiera's tiled position embedding behavior prevents it from interpolating properly. But instead of treating it as a problem, we fix it by *embracing* the

tiling behavior and *baking it in* to the position embeddings. Since Hiera has both window and global attention (window in stages 1 and 2, global in 3 and 4), we define a separate *window embedding* and *global embedding*. The former is the size of one window ($8 \times 8$ tokens), while the latter is the size of the feature map without windows ($7 \times 7$). Both are randomly initialized and learned during training.

To construct Hiera's full $56 \times 56$ position embedding, we *tile* the window embedding and *interpolate* the global embedding, then add the results (see Fig. 3). This is similar to the learned behavior of absolute embeddings (Fig. 2), but we can now interpolate properly (Fig. 1e): tile the window embedding and interpolate the global embedding to the desired size (cropping as necessary). This is a drop in replacement to the original absolute position embeddings, but with slightly fewer parameters.

To test our strategy, we pretrain a Hiera-L model with 400 epochs of MAE on ImageNet-1k using 224px images and then finetune on either 224px or 256px images in Tab. 1. When finetuning on 256px images, the original absolute embeddings *drop* accuracy by 0.4%. Our absolute window embeddings, on the other hand, both *increase* accuracy at 224px (slightly, likely due to faster convergence) *and* obtain the expected accuracy after finetuning at 256px. Moreover, they *significantly increase* performance for a fully supervised Hiera-B.[1] Thus, we entitle this method "absolute win".

**Analysis.** We can study the effect position embeddings have by measuring their *similarity*. If the position embedding for two tokens are similar, then they are likely to be paired in attention (see Appendix E). We present the similarity of individual tokens within the learned $8 \times 8$ window and $7 \times 7$ global embeddings in Fig. 5 for our Hiera-L model trained with absolute win. The global embeddings act as expected: tokens close by (or in a close row or column) are considered similar. However, the window embeddings act very differently. In fact, they seem to perform *dilated convolution*.

Furthermore, we study the importance of each component in Tab. 2. Our default $7 \times 7$ size for global embeddings seems to be optimal. Moreover, both window and global embeddings alone fix the bug, since the window embed will always align with window attn and the global embed is at the resolution where windows are pooled to $1 \times 1$. Nevertheless, together they result in the most performant model.

## 4    EXPANDING TO DETECTION

In Sec. 3, we explored how an interaction between window attention and absolute position embeddings caused undesirable behavior in learned position embeddings using Hiera (Ryali et al., 2023) as our test platform. However, Hiera is not the only model that uses both window attention and absolute position embeddings. In fact, a common strategy when applying vanilla vision transformers (ViT, Dosovitskiy et al. (2020)) to high resolution downstream tasks is to *add* window attention.

### 4.1    VITDET

Of particular note is ViTDet (Li et al., 2022a), a popular method to use ViTs for object detection. ViTDet takes a standard 224px ViT model (i.e., with $14 \times 14$ tokens) pretrained with MAE (He et al., 2022), and then finetunes at resolutions of 1024px or more. But doing this with global attention is too expensive, so instead the authors convert all but 4 of the attention layers in the model to window attention, with each window being $14 \times 14$ tokens (i.e., the same as the pretrained model). Naturally, ViT uses absolute position embeddings, so to allow the model to input larger images, ViTDet naïvely interpolates them. Finally, ViTDet adds *relative position embeddings* (Li et al., 2022b) to the attention module in each layer, which the authors find important for good performance.

**Does ViTDet also suffer from the bug?** ViTDet has all the necessary components for the titular bug to occur: window attention, absolute position embeddings, and naïve interpolation; but this time, in a different order. Hiera trains a model with both absolute position embeddings and window attention, and then tries to interpolate the result. On the other hand, ViTDet takes an interpolatable position embedding (learned or sinusoidal), interpolates it, and *then* adds window attention during finetuning.

While not as egregious, this is still wrong. As described in Sec. 3, position embeddings are how attention implements spatial operations. Drastically changing the position embeddings when finetuning, then, would change the operation performed from what the model learned during pretraining.

---

[1]This suggests pure transformers (ViT, Hiera) learn suboptimal pos embeds when fully supervised on IN-1k. With careful design, it may be possible to obtain MAE-like performance w/ supervised training (see Appendix B).

| relpos | abs win | COCO mAP box | mask | ms/im |
|--------|---------|-----|------|-------|
| all | ✗ | 55.6 | 49.2 | 97 |
|  | ✓ | **55.8** (+0.2) | 49.3 (+0.1) | 97 |
| none | ✗ | 54.6 | 48.5 | 66 |
|  | ✓ | 55.0 (+0.4) | 49.1 (+0.6) | 66 |
| ga only | ✗ | 55.1 | 48.9 | 69 |
|  | ✓ | **55.8** (+0.7) | **49.6** (+0.7) | 68 |

Table 3: **Fixing ViTDet-L** by tiling the original position embeddings. Absolute win reduces the need for relative position embeddings (relpos) to only the 4 global attention (ga) layers.

| strategy | location | COCO mAP box | mask | ms/im |
|----------|----------|-----|------|-------|
| relative | all attn | **55.8** | 49.3 | 97 |
|  | win attn | 55.0 | 49.0 | 75 |
|  | global attn | **55.8** | **49.6** | 68 |
| absolute | patchify | 55.1 | 49.1 | 66 |
|  | global block | 55.1 | 49.0 | 66 |
|  | global qkv | 55.0 | 49.0 | 67 |

Table 4: **Global Embed for ViTDet.** Learning absolute position embeddings from scratch during finetuning doesn't work, so we opt to use relpos added to just global attention instead.

Interpolating the position embedding and adding window attention certainly "drastically changes the position embeddings" for each window by splitting one embedding into multiple parts (see Fig. 1cd).

**Applying Absolute Win.** To fix this, we can apply the same "absolute win" technique, but in a different way. The original ViT model operates at a resolution of $14 \times 14$ tokens. Based on that, ViTDet sets the size of each window to $14 \times 14$ in the interpolated model. Thus, if we want attention to perform the same operation it did during pretraining, we simply need to ensure that each $14 \times 14$ window has the same position embedding—i.e., by *tiling* the position embeddings instead of naïvely interpolating them. Essentially, we construct an absolute win embedding with the original pretrained position embeddings acting as window embeddings (Fig. 6). Note that ViTDet has 4 global attention layers, so we still need a global embedding. We will discuss how to achieve that in the next section.

If we apply this technique to a ViTDet model on COCO (Lin et al., 2014) using the same training setup as Li et al. (2022a), we only get small gains for both box and mask (see Tab. 3). Does this mean that ViTDet is not actually affected by the bug? Actually, the secret is in the relative position embeddings (relpos): not adding them during finetuning drops performance by up to 1 mAP box and 0.7 mAP mask for the original model. If we then apply absolute win *without relpos*, we gain +0.4 mAP box and +0.6 mAP mask. This means that a portion of the gain for box mAP and most of the the gain from mask mAP for using relpos in the first place is due to the bug.

**Removing Relative Position Embeddings.** Does this matter? If relpos can make up the difference, why would we want to remove it? Simple: relpos is *slow*. In Tab. 3, we present the inference runtime for each model (A100 fp16). Using relpos in every attention layer can slow the model down by *47%*.

We now return to the component missing from our implementation of absolute win for detection: the global embedding. Using the pretrained model's embeddings as the finetuned model's window embeddings is correct *for the window attention layers*. However, the model still has 4 global attention layers that now use the wrong position embeddings. We addressed this in Sec. 3 with a global embedding, but the pretrained model does not have one here. We could try to train our own during finetuning, but this would adversely affect the windows from pretraining (Tab. 4: absolute, patchify).

Instead, we want to inject this position information directly into the layers with global attention, avoiding those with window attention. However, if we add a zero-initialized absolute position embedding to either the input to the block or the outputs of qkv (as is done in DETR (Carion et al., 2020)) for blocks with global attention, we also see no gain (Tab. 4). Not being able to learn good absolute embeddings during supervised finetuning is not so surprising, as we found supervised Hiera-B in Sec. 3 to also not be able to learn good absolute position embeddings from scratch. But we *do* have an embedding strategy that does not require a good pretext task to learn: namely, relpos.

Thus, our final method for absolute win applied to ViTDet is to *tile* the pretrained model's position embeddings and learn new relative position embeddings *for only the 4 global attention layers*. The result in Tab. 3 is a model that is *stronger* than the original ViTDet-L, while *being significantly faster*.

## 4.2 HIERADET

If absolute win can improve ViT for detection, then what about Hiera? In Ryali et al. (2023), Hiera outperforms ViT in almost all cases when not finetuned at a higher res. However, Hiera-L performs

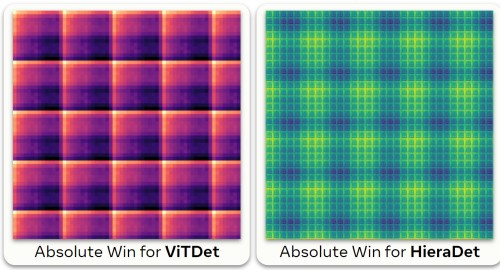

| | Absolute Win for **ViTDet** | Absolute Win for **HieraDet** |

Figure 6: **Absolute Win for Detection.** For detection, we apply absolute win *recursively* by tiling the original model's pos embed.

| model | abs win | COCO mAP box | mask | ms/im |
|---|---|---|---|---|
| ViT-L | ✗ | 55.6 | 49.2 | 97 |
| ViT-L | ✓ | 55.8 | 49.6 | 68 |
| Hiera-L | ✗ | 55.0 | 48.6 | - |
| Hiera-L‡ | ✗ | 55.6 | 49.3 | 92 |
| Hiera-L‡ | ✓ | **56.2** | **50.0** | **67** |

Table 5: **Fixing HieraDet-L** by applying absolute win *recursively*. Our implementation (‡) already outperforms the original (Ryali et al., 2023), but absolute win embeddings allows HieraDet to considerably outperform even ViTDet.

*significantly worse* than ViT-L for detection (55.0 v.s. 55.6 box mAP on COCO). Of course, this result is bugged. Hiera uses the same set up as ViTDet, i.e., funetune at 1024px images. Thus, we should expect to see a drop in accuracy as in Tab. 1. Naturally, we can fix this with absolute win.

**Building a Model.** We build on the methodology of Ryali et al. (2023) to construct our HieraDet. We replace the ViT backbone with Hiera and use an FPN (Lin et al., 2017) to aggregate multi-scale features. We then use the same Mask R-CNN (He et al., 2017) heads as ViTDet. Similar to ViTDet, we replace the global attention in Hiera with window attention *of the same scale* (i.e., $14 \times 14$ windows for stage 3 and $7 \times 7$ windows for stage 4) and leave the existing windows in stages 1 and 2 unchanged. Then to allow mixing between windows like ViTDet, we choose 3 equally dispersed attention layers in stage 3 to remain global (which we find to be optimal, see Appendix C).

**Applying Absolute Win.** But what do we do for position embeddings? We now have two "scales" of window attention: the $8 \times 8$ windows originally within Hiera, and the $56 \times 56$ windows (i.e., $14 \times 14$ at stage 3 and $7 \times 7$ at stage 4 after downsampling) added following ViTDet. Simple—we have multiple *scales* of window attention, so apply absolute win *recursively*. At the lowest level, we have our pretrained Hiera model with $8 \times 8$ window and $7 \times 7$ global embeddings. Then we construct a $56 \times 56$ embedding from these as in Fig. 3, and then use *that* as the window embedding for the entire HieraDet model (see Fig. 6). We also add relpos to just the 3 global attention layers to serve as the new global embedding. Finally, with the position embeddings corrected, we can now use layer-wise lr decay (Clark et al., 2020; Bao et al., 2021) which boosts performance for larger models.[2]

We show the results of these changes in Tab. 5 for a Hiera-L on COCO. Without absolute win embeddings, our HieraDet is already +0.6 box mAP and +0.7 mask mAP over the original Hiera for detection results in Ryali et al. (2023), matching the performance of ViTDet-L. However, applying absolute win unlocks Hiera's performance even further, adding *another +0.6 box mAP and +0.7 mask mAP on top*. This makes our HieraDet-L with absolute win **+1.2 box mAP** and **+1.4 mask mAP** over the original, *while being faster* (due to having only 3 relpos).

## 5 RESULTS

With the bug fixed, both Hiera and ViT perform better on several tasks. Here we evaluate absolute win v.s. the original models and other state-of-the-art. See Appendix A for implementation details.

**Image Recognition.** In Tab. 6 we evaluate the effectiveness of absolute win for Hiera on ImageNet-1k (Deng et al., 2009) by pretraining for 1600 epochs on 224px images[3] using the hyperparameters from Ryali et al. (2023) and then finetuning on either 224px or 384px images. We then compare the resulting models to other existing work, i.e. MViTv2 (Li et al., 2022b) and Swin (Liu et al., 2021b), that have both 224px and 384px results as well as the original Hiera models (Ryali et al., 2023) which we finetune at 384px. In particular, we care about the *performance increase* from finetuning at a larger image size (i.e., not the raw acc, as that would be unfair for supervised models). Compared to models of the same compute, Hiera with absolute win benefits *more* from increased resolution,

---

[2]ViTDet uses this, but the Hiera detection experiments in Ryali et al. (2023) do not. It turns out that this is because the bugged interpolated embeddings need to be retrained in order to get good accuracy.

[3]See Appendix F for a discussion about pretraining with higher resolutions.

| model | 224px acc | flops | 384px acc | flops | Δ acc |
|---|---|---|---|---|---|
| MViTv2-B† | 84.4 | 10.2G | 85.2 | 36.7G | +0.8 |
| Hiera-B | **84.5** | 9.4G | 85.0 | 30.3G | +0.5 |
| **Hiera-B**_abs win_ | **84.5** | 9.4G | **85.8** | 30.3G | **+1.3** |
| Swin-B† | 83.5 | 15.4G | 84.5 | 47.0G | +1.0 |
| Hiera-B+ | **85.2** | 12.7G | 85.2 | 40.3G | +0.0 |
| **Hiera-B+**_abs win_ | 85.1 | 12.7G | **86.2** | 40.3G | **+1.1** |
| MViTv2-L† | 85.3 | 42.1G | 86.0 | 140G | +0.7 |
| Hiera-L | **86.1** | 40.3G | 86.4 | 128G | +0.3 |
| **Hiera-L**_abs win_ | **86.1** | 40.3G | **86.9** | 128G | **+0.8** |
| Hiera-H | **86.9** | 125G | 86.7 | 383G | -0.2 |
| **Hiera-H**_abs win_ | 86.8 | 125G | **87.3** | 383G | **+0.5** |

Table 6: **ImageNet-1k Results.** Comparing to other models that train at both 224px and 384px. Hiera with our absolute win gains more from high res than others (Li et al., 2022b; Liu et al., 2021b; Ryali et al., 2023). † Fully supervised.

| model | in-1k acc @ ft res 224px | 256px | 320px | 448px |
|---|---|---|---|---|
| Hiera-B | 84.5 | 84.2 (-0.3) | 84.6 (+0.1) | 84.8 (+0.3) |
| **w/ abs win** | 84.5 | **84.9** (+0.4) | **85.4** (+0.9) | **85.7** (+1.2) |
| Hiera-B+ | 85.2 | 84.3 (-0.9) | 85.1 (-0.1) | 84.9 (-0.3) |
| **w/ abs win** | 85.1 | **85.3** (+0.2) | **86.0** (+0.9) | **86.2** (+1.1) |
| Hiera-L | 86.1 | 85.7 (-0.4) | 86.2 (+0.1) | 86.6 (+0.5) |
| **w/ abs win** | 86.1 | **86.5** (+0.4) | **86.7** (+0.6) | **87.0** (+0.9) |
| Hiera-H | 86.9 | 86.5 (-0.4) | 86.9 (-0.0) | 86.5 (-0.4) |
| **w/ abs win** | 86.8 | **87.0** (+0.2) | **87.2** (+0.4) | **87.3** (+0.5) |

Table 8: **Image Size Comparison.** Evaluating the performance of absolute win on additional finetune resolutions. Without absolute win, performance significantly degrades. Note that we use the same hyperparameters in each experiment here, which is not necessarily optimal.

| model | 16×224px acc | flops | 32×320px acc | flops | Δ acc |
|---|---|---|---|---|---|
| ViT-L | 85.2 | 597G | 86.1 | 3958G | +0.9 |
| Hiera-L | **87.3** | 413G | 87.4 | 2962G | +0.1 |
| **Hiera-L**_abs win_ | 87.1 | 413G | **88.0** | 2962G | **+0.9** |
| ViT-H | 86.6 | 1192G | 87.4 | 7397G | +0.8 |
| Hiera-H | **87.8** | 1159G | 88.1 | 7003G | +0.3 |
| **Hiera-H**_abs win_ | 87.6 | 1159G | **88.4** | 7003G | **+0.8** |

Table 7: **Kinetics-400 Results.** Frames × img size w/ 3 spatial and 5 temporal crops. ViT uses 4 temporal crops at high res (Tong et al., 2022).

| model | frames | size | mAP | flops |
|---|---|---|---|---|
| ViT-L | 16 | $224^2$ | 37.0 | 597G |
| MViTv2-L | 40 | $312^2$ | 38.5 | 2828G |
| Hiera-L | 16 | $224^2$ | 39.8 | 413G |
| **Hiera-L**_abs win_ | 32 | $320^2$ | **42.4** | 2962G |
| ViT-H | 16 | $224^2$ | 39.5 | 633G |
| Hiera-H | 16 | $224^2$ | 42.5 | 672G |
| **Hiera-H**_abs win_ | 32 | $320^2$ | **43.8** | 7003G |

Table 9: **AVA v2.2 Results.** Video action localization compared to models pretrained on K400 with masked image modeling (Tong et al., 2022; Ryali et al., 2023; Wei et al., 2022).

while retaining 224px acc. We further test this in Tab. 8, where we present more finetune resolutions. While the original model can lose accuracy at some resolutions, absolute win solves the issue.

**Video Recognition.** We perform a similar experiment in Tab. 7 on Kinetics-400 (Kay et al., 2017) action classification for video using 3200 ep of pretraining and the same hyperparameters as Ryali et al. (2023). We then finetune using 32 frames (each of 320px) to compare with Video MAE (Tong et al., 2022). We interpolate the temporal embedding linearly and use absolute win on the spatial embedding. While our 224px models are not as strong as Hiera's, our 320px results clearly show the benefits of absolute win, resulting in a Hiera-L model that outperforms Video MAE by **1-2%** at high res. In Tab. 9, we test this further by finetuning the resulting model on AVAv2.2 (Gu et al., 2018), an action localization dataset. This allows us to outperform the prior state-of-the-art for K400 pretraining by **+2.6** AP for L and **+1.3** AP for H. Thus, absolute win is as effective on video as images.

**Object Detection and Segmentation.** We evaluate our absolute win applied to ViTDet and HieraDet for object detection and instance segmentation on COCO (Lin et al., 2014) using Detectron2 (Wu et al., 2019) training on train2017 and testing on val2017. ViT baselines are from ViTDet (Li et al., 2022a) and Hiera baselines are from the original paper (Ryali et al., 2023). For ViT, we keep all hyperparameters the same, but for Hiera we use our own implementation as described in Sec. 4.2.

In Tab. 10, we compare ViTDet and HieraDet with absolute win the results from their original papers by finetuning Mask R-CNN (He et al., 2017) or Cascade Mask R-CNN (Cai & Vasconcelos, 2019). In every case, absolute win improves performance. Notably, the performance increase is slight for ViTDet (as its extra relpos hides the improvement), but our version is much faster (see Fig. 7). Furthermore, absolute win both speeds up HieraDet and results in huge performance improvements: up to **1.5 AP^box** and **1.6 AP^mask** for the base model. The resulting Hiera models outperform ViTDet.

In Fig. 7, we benchmark our models against ViTDet as well as MViTv2 (Li et al., 2022b) and Swin (Liu et al., 2021b) using ImageNet-21k pretraining from Li et al. (2022a), and the more recent Swin

| model | Mask R-CNN | | Cascade Mask R-CNN | |
|---|---|---|---|---|
| | $AP^{box}$ | $AP^{mask}$ | $AP^{box}$ | $AP^{mask}$ |
| ViT-B | 51.6 | 45.9 | 54.0 | 46.7 |
| **ViT-B**$_{abs\ win}$ | **52.2** | **46.2** | **54.1** | **47.0** |
| Hiera-B | 52.2 | 46.3 | - | - |
| **Hiera-B**$^{‡}_{abs\ win}$ | **53.7** | **47.9** | **55.8** | **48.4** |
| Hiera-B+ | 53.5 | 47.3 | - | - |
| **Hiera-B+**$^{‡}_{abs\ win}$ | **54.3** | **48.2** | **56.8** | **49.1** |
| ViT-L | 55.6 | 49.2 | 57.6 | 49.8 |
| **ViT-L**$_{abs\ win}$ | **55.8** | **49.6** | **57.8** | **50.0** |
| Hiera-L | 55.0 | 48.6 | - | - |
| **Hiera-L**$^{‡}_{abs\ win}$ | **56.2** | **50.0** | **58.4** | **50.7** |
| ViT-H | 56.7 | 50.1 | 58.7 | 50.9 |
| **ViT-H**$_{abs\ win}$ | **56.9** | **50.3** | **58.8** | **51.0** |
| **Hiera-H**$^{‡}_{abs\ win}$ | **57.0** | **50.3** | **59.0** | **51.1** |

Table 10: **COCO w/ Absolute Win.** Comparison between our ViTDet and HieraDet with absolute win and without. $^{‡}$ Our implementation.

| model | single-scale test | | multi-scale test | |
|---|---|---|---|---|
| | $AP^{box}$ | $AP^{mask}$ | $AP^{box}$ | $AP^{mask}$ |
| Swin-L $_{HTC++,\ 21K,\ sup}$ | 57.1 | 49.5 | 58.0 | 50.4 |
| MViTv2-H $_{Cas,\ 21K,\ sup}$ | 57.1 | 48.8 | 58.4 | 50.1 |
| CBNetV2 $_{HTC,\ 21K,\ sup}$ | 59.1 | 51.0 | 59.6 | 51.8 |
| ViTDet-H $_{Cas,\ 1K,\ MAE}$ | 60.4 | **52.0** | 61.3 | 53.1 |
| *w/ absolute win* | | | | |
| **Hiera-B** $_{Cas,\ 1K,\ MAE}$ | 57.7 | 49.6 | 59.3 | 51.2 |
| **Hiera-B+** $_{Cas,\ 1K,\ MAE}$ | 58.5 | 50.4 | 60.0 | 52.0 |
| **Hiera-L** $_{Cas,\ 1K,\ MAE}$ | 60.1 | 51.6 | 61.5 | 53.0 |
| **Hiera-H** $_{Cas,\ 1K,\ MAE}$ | **60.7** | **52.0** | **61.7** | **53.3** |

Table 11: **COCO Results** on minival for methods pretrained on ImageNet (Liu et al., 2021b; Li et al., 2022b; Liang et al., 2022; Li et al., 2022a).

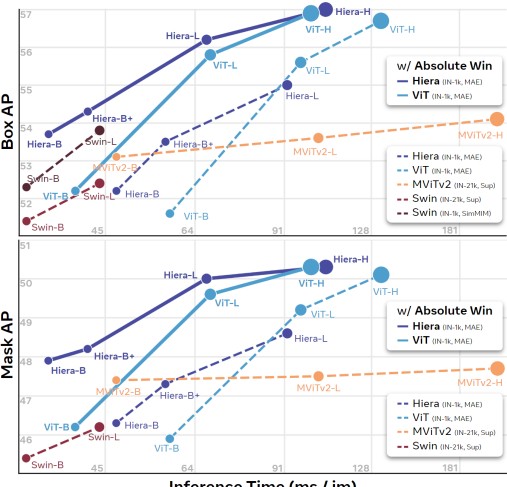

Figure 7: **COCO Runtime.** Benchmarked on the same A100 using Mask R-CNN (fp16, bs=8).

| model | single-scale test | | |
|---|---|---|---|
| | $AP^{mask}$ | $AP^{mask}_{rare}$ | $AP^{box}$ |
| Detic $_{Zhou\ et\ al.\ (2022)}$ | 41.7 | 41.7 | - |
| CBNetV2 $^{2021\ Comp\ Winner}_{Fu\ et\ al.\ (2021)}$ | **49.2** | **45.4** | - |
| ViTDet-L $_{Li\ et\ al.\ (2022a)}$ | 46.0 | 34.3 | 51.2 |
| ViTDet-H $_{Li\ et\ al.\ (2022a)}$ | 48.1 | 36.9 | 53.4 |
| *w/ absolute win* | | | |
| **Hiera-B** $_{abs\ win}$ | 42.9 | 32.0 | 47.7 |
| **Hiera-B+** $_{abs\ win}$ | 43.9 | 30.7 | 48.9 |
| **Hiera-L** $_{abs\ win}$ | 47.1 | 37.4 | 52.5 |
| **Hiera-H** $_{abs\ win}$ | 48.9 | 38.0 | **54.6** |

Table 12: **LVIS v1 Results.** System-level comparison using the same setup as Li et al. (2022a).

using SimMIM (Xie et al., 2022) pretraining. Our method significantly speeds up both ViTDet and HieraDet, with the resulting HieraDet being faster and more accurate than everything else.

Finally, like in Li et al. (2022a), we present additional results using cascade (cas), 1280px images, and softnms (Bodla et al., 2017) with and without test-time augmentation. In Tab. 11, we compare the resulting model to other ImageNet pretrained work. HieraDet with absolute win is *extremely* strong: with our bug fix, **Hiera-B** outperforms methods using heavier backbones and pipelines like HTC (Chen et al., 2019). Moreover, multi-scale testing requires interpolating the position embedding. Thus, our models with absolute win benefit more, with **Hiera-L** outperforming **ViT-H** on box mAP.

We conduct the same experiment on LVIS (Gupta et al., 2019) using the setup from ViTDet in Tab. 12. Our version of HieraDet also significantly outperforms ViTDet. It comes close to the results from the LVIS-optimized 2021 challenge winner (Fu et al., 2021), despite using only baseline LVIS strategies.

## 6 CONCLUSION

We find a bug, fix it with absolute win, and show extensively how doing so improves the state-of-the-art. Our findings lead to a few practical suggestions. From Hiera: embracing the emergent behavior of a model can lead to significant gains. From ViTDet: minimizing the disparity between pretraining and finetuning can lead to optimal performance. But more than that, we discover a potential issue for any method with window attention and absolute position embeddings. While we only show two examples here, several others use these components. Moreover, other methods could be partially affected—e.g., Swin has window attention but uses relative embeddings, which if replaced with absolute embeddings could lead to a faster model. We hope that future work can use these techniques to obtain a more powerful and more efficient state-of-the-art, which we would see as an *absolute win*.

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

# A    IMPLEMENTATION DETAILS

Here we list the details and configurations for each set of experiments.

## A.1    IMAGE RECOGNITION

For our image recognition experiments, we train and evaluate on ImageNet-1k (Deng et al., 2009) using MAE pretraining for 1600 epochs following the same hyperparameters in Ryali et al. (2023). For finetuning, our hyperparameters follow Ryali et al. (2023) for the most part. We us these same hyperparameters for all image sizes in Tab. 6 and Tab. 8. Note that different hyperparameters might be optimal for higher resolution, but that is out of scope for our experiments here. We list the settings we use for finetuning in Tab. 13 below.

| config | value |
|---|---|
| optimizer | AdamW |
| optimizer momentum | $\beta_1, \beta_2 = 0.9, 0.999$ |
| weight decay | 0.05 |
| learning rate | 2e-3 (B); 1e-3 (B+, L, H) |
| learning rate schedule | cosine decay |
| warmup epochs | 5 |
| epochs | 100 (B, B+); 50 (L, H) |
| augmentation | RandAug (9, 0.5) (Cubuk et al., 2020) |
| batch size | 1024 |
| mixup (Zhang et al., 2018) | 0.8 |
| cutmix (Yun et al., 2019) | 1.0 |
| label smoothing (Szegedy et al., 2016) | 0.1 |
| drop path (Larsson et al., 2017) | 0.1 (B, B+); 0.2 (L); 0.3 (H) |
| dropout (Srivastava et al., 2014) | 0.7 (B, B+); 0.9 (L); 0.85 (H) |

Table 13: **Finetuning settings for ImageNet-1K.** We use the same settings for each image size.

Note that we finetune the large image models directly from the pretrained model. Also, for larger image sizes for H models, we had to use both activation checkpointing and torch 2.0 scaled dot product attention to allow the same batch size to fit in GPU memory. The latter could affect accuracy.

## A.2    VIDEO RECOGNITION

For video recognition, we conduct experiments on Kinetics-400 (Kay et al., 2017) action classification and AVA v2.2 (Gu et al., 2018) action localization. Like for ImageNet-1k, for Kinetics-400, we use the same 3200 epoch pretraining settings as Ryali et al. (2023), as well as the same finetuning settings for the original $16 \times 224$px resolution. For $32 \times 320$px, we finetune with the settings in Tab. 14.

| config | value |
|---|---|
| optimizer | AdamW |
| optimizer momentum | $\beta_1, \beta_2 = 0.9, 0.999$ |
| weight decay | 1e-8 |
| learning rate | 1.6e-4 |
| learning rate schedule | cosine decay |
| epochs | 30 |
| repeated sampling | 2 |
| augmentation | RandAug (7, 0.5) (Cubuk et al., 2020) |
| batch size | 128 |
| gradient clipping | 5.0 |
| label smoothing (Szegedy et al., 2016) | 0.1 |
| drop path (Larsson et al., 2017) | 0.5 ($L_{32 \times 320px}$), 0.6 ($H_{32 \times 320px}$) |
| dropout (Srivastava et al., 2014) | 0.5 |
| layer-wise decay (Clark et al., 2020) | 0.9 |

Table 14: **Finetuning settings for Kinetics-400.** Finetuning at 32 frames and 320px resolution.

We finetune the high res models *on top* of the $16 \times 224$px finetuned model, not from the pretrained one (to save time). Activation checkpointing was used for both models, and mixup was disabled because

each gpu had a batch size of 1. To interpolate the position embeddings temporally, we linearly interpolate the temporal embeddings from Hiera, and use absolute win for the spatial embeddings. Like with the low resolution models, we sample frames with a temporal stride of 4.

We take the 32×320px finetuned Kinetics-400 model and finetune it further on AVA v2.2 using the following settings in Tab. 15.

| config | values |
|---|---|
| optimizer | SGD |
| weight decay | 1e-8 |
| learning rate | 3.6 ($L_{32\times320px}$), 3.2 ($H_{32\times320px}$) |
| learning rate schedule | cosine decay |
| warmup epochs | 5 ($L_{32\times320px}$), 8 ($H_{32\times320px}$) |
| epochs | 30 |
| batch size | 128 |
| drop path (Larsson et al., 2017) | 0.5 |
| dropout (Srivastava et al., 2014) | 0.5 |
| layer-wise decay (Clark et al., 2020) | 0.9 |

Table 15: **Finetuning settings for AVA.** Finetuning at 32 frames and 320px resolution.

The H model required us to use a batch size of 1 per GPU even with activation checkpointing, which we found to be unstable. Thus, we use a lower learning rate and more warmup epochs for the H model to compensate. It is possible that a higher learning rate would result in better performance.

## A.3 DETECTION AND INSTANCE SEGMENTATION

For detection with ViTDet, we use all the same parameters as Li et al. (2022a). Our settings for HieraDet, however, are very different to Ryali et al. (2023). We list those settings in Tab. 16 here:

| config | values |
|---|---|
| optimizer | AdamW |
| optimizer momentum | $\beta_1, \beta_2 = 0.9, 0.999$ |
| weight decay | 0.1 |
| learning rate | 1.4e-4 (B); 7e-5 (B+); 1.2e-4 (L); 6e-5 (H) |
| learning rate schedule | step-wise decay |
| epochs | 100 (B, B+, L); 75 (H) |
| augmentation | LSJ [0.1, 2.0] |
| batch size | 64 |
| drop path (Larsson et al., 2017) | 0.2 (B); 0.3 (B+); 0.45 (L); 0.55 (H) |
| layer-wise decay (Clark et al., 2020) | 0.85 (B, B+); 0.9 (L); 0.925 (H) |
| window attn size | 8, 4, 14, 7 |
| global attn layers | 12-16-20 (B, B+); 23-33-43 (L, H) |
| relpos | global attn layers only |

Table 16: **Settings for COCO.** HieraDet finetuning settings for both Mask and Cascade R-CNN.

We use these settings for every COCO experiment, except for our 1280px experiment where we increase the droppath for H to 0.6. When we use softnms (Bodla et al., 2017), we use the linear method.

Our settings for LVIS are the same as in Tab. 16, except we use repeat factor sampling and federated loss as in Li et al. (2022a). To compensate we also adjust the learning rates: 1.4e-4 (B); 1.4e-4 (B+); 2.4e-4 (L); 6e-5 (H). Finally, all models use 100 epochs for LVIS like in Li et al. (2022a).

## A.4 SPEED BENCHMARKING

We use a single NVIDIA A100 40GB GPU to benchmark speed for all baselines and our approach. All detection results were benchmarked with a batch size of 8 using fp16 in detectron2, taking multiple tests and throwing out the first 25%. For attention layers (in ViT and Hiera) *not* using relpos, we accelerate attention using torch 2.0's scaled dot product attention function. We used Li et al.'s Detectron2 code for each approach. We approximate Hiera without absolute win as our version of HieraDet with all layers having relpos. The full benchmark results are as follows in Tab. 17.

|        |    | ms / im |     |     |
|--------|----|---------|-----|-----|
| method | B  | B+      | L   | H   |
| MViTv2 | 47 | -       | 104 | 210 |
| Swin   | 33 | -       | 44  | -   |
| ViT    | 58 | -       | 97  | 133 |
| **ViT** abs win | 40 | - | 68 | 101 |
| Hiera  | 47 | 57      | 92  | -   |
| **Hiera** abs win | 36 | 42 | 67 | 107 |

Table 17: **Detection Benchmarking.** The numbers for Fig. 7. Using 1024px images on COCO with Mask R-CNN on a single A100 with a batch size of 8 and half precision.

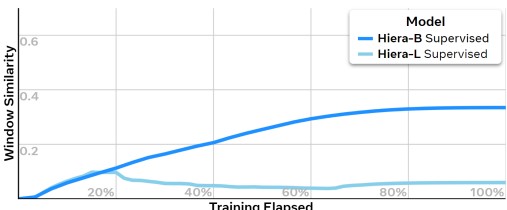

Figure 8: **Window Repetition** for *fully supervised* baseline Hiera models on ImageNet-1k. Like Fig. 4, but here showing both Hiera-B and Hiera-L models. Notably, the L model does not reach the same level of window similarity as B.

| model | pos embed | acc | $\Delta$ |
|-------|-----------|-----|----------|
| Hiera-L$_{Sup}$ | absolute | 76.1 | |
|  | absolute win | **79.8** | **+3.7** |
| Hiera-B$_{Sup}$ | absolute | 81.8 | |
|  | absolute win | **82.4** | **+0.6** |

Table 18: **Fully Supervised Training.** Absolute win can significantly improve the performance of a Hiera model trained from scratch. However, Hiera-L here benefits significantly more than Hiera-B. Fig. 8 has a potential answer: Hiera-L doesn't learn the repeated window pattern.

## B    ABSOLUTE WIN FOR FULLY SUPERVISED MODELS

It's common knowledge that training pure transformers fully supervised with only ImageNet-1k data is *suboptimal* (Dosovitskiy et al., 2020; Zhai et al., 2021; He et al., 2022). This is true for Hiera as well (Ryali et al. (2023) Appendix). However, the specific mechanism that causes this deficiency or the details behind why pretraining with, e.g., MAE helps so much is still not fully understood.

In Sec. 3, make two interesting observations: Hiera's window repetition behavior is less pronounced when training fully supervised from scratch than when training with MAE, and absolute win can improve performance for fully supervised models. In order to shed some light on the differences between what supervised and MAE-pretrained models learn, we explore these two observations.

In Fig. 8, we show the window similarity over fully supervised training of a Hiera model from scratch. While the Hiera-B model learns some kind of repeated window embedding, the Hiera-L model specifically *does not*. And we find that this is correlated with accuracy. In Tab. 18, we show the final accuracies of these two models, and the Hiera-L model performs significantly worse than the Hiera-B model. This is not surprising, since it is likely for these larger models to overfit on ImageNet-1k. However, interestingly using absolute win *increases the accuracy of the L model by 3.7%*.

We would like to emphasize: *simply rearranging a few parameters **at the start of the network** drastically improves performance for a supervised Hiera-L model.* It does not seem reasonable, then, that the accuracy drop is a wholly attributable to overfitting. There are, after all, 48 layers of parameters between the position embeddings and the outputs of a Hiera-L model. If the model can overfit with one position embedding, then it can overfit with a slightly different position embedding.

Instead of overfitting, it seems that these models do not learn a good prior for attention. While MAE requires spatial reasoning to perform well, fully supervised classification does not. Attention does not have explicit signal from supervised classification, making it more likely for a fully supervised model to get stuck in a bad local optimum for attention. Simply rearranging the position embedding can allow the model to find a new optimum that makes better use of spatial information.

But, we note that supervised training is not necessarily the problem. To test this, we perform one more experiment through the lens of repeated window position embeddings. This time, we take a baseline Hiera-L model pretrained with MAE and *reset* the position embeddings, then do 50 epochs of supervised training. In Fig. 9, we compare this to training that same model from scratch (for 300 supervised epochs). And interestingly enough, *the model is able to relearn its position embeddings*

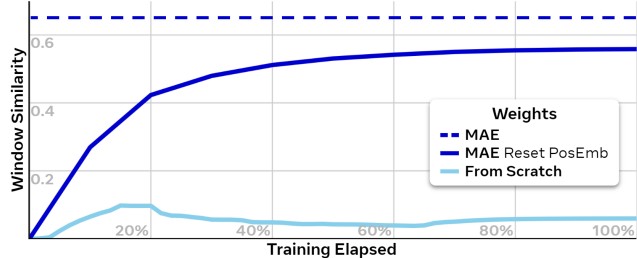

Figure 9: **Resetting the Pos Embed** for an MAE pretrained Hiera-L model before finetuning. Unlike training completely from scratch, the MAE-pretrained model quickly relearns a position embedding with high window similarity. However, it doesn't fully reach the original similarity.

*under supervised training*. Thus, the task itself is not directly responsible for the repeated position embeddings. Instead, the learned attention operation is, meaning that MAE enables the model to settle into a good local optimum for attention. Though, note that the model with reset position embeddings does not relearn exactly the same position embedding (falling just short on window similarity). It obtains 84.1% accuracy v.s. the original model's 85.6%.

A couple of take-aways from this line of inquiry: (1) Overfitting may not be the full story of poor performance of large transformers trained from scratch on ImageNet-1k - learning *suboptimal* attention operations may play a role; (2) the improved results from MAE pretraining can potentially be replicated when training with only supervised classification, provided the model finds the right local optimum; (3) absolute win, or some other way of subtly nudging the model in the right direction, can significantly improve performance for fully supervised models. We leave further exploration of this subject to future work.

## C  HIERADET ABLATIONS

Here we ablate the global attention placement for HieraDet mentioned in Sec. 4.2. As described in Tab. 19 below, we find that placing global attn layers too early or too late in the network hurts performance. Then, we find that we only need 3 global attention layers to get optimal performance. Finally, we ablate the stride for placing these layers and find 10 to be optimal. So, our final placement for Hiera-L is the last layer of stage 3 (43) and then two more layers placed before it at 33 and 23.

| ga layers | $AP^{box}$ | $AP^{mask}$ |
|---|---|---|
| none | 55.2 | 49.2 |
| $10_{s3}$ | 54.0 | 48.2 |
| $10_{s3}, 26_{s3}$ | 55.2 | 49.2 |
| $26_{s3}, 42_{s3}$ | **55.8** | **49.6** |

(a) **Early Layers.** Placing a ga layer early (e.g., layer 10) actually *hurts* performance.

| ga layers | $AP^{box}$ | $AP^{mask}$ |
|---|---|---|
| none | 55.2 | 49.2 |
| $42_{s3}$ | **55.4** | **49.4** |
| $42_{s3}, 47_{s4}$ | 55.3 | 49.2 |

(b) **Late Layers.** Making the last layer in stage 4 a ga layer also hurts performance slightly.

| ga layers | $AP^{box}$ | $AP^{mask}$ |
|---|---|---|
| $31_{s3}, 43_{s3}$ | 55.9 | 49.6 |
| $19_{s3}, 31_{s3}, 43_{s3}$ | **56.0** | **49.9** |
| $19_{s3}, 27_{s3}, 35_{s3}, 43_{s3}$ | **56.0** | 49.7 |

(c) **Number of Layers.** We do not find any benefit from using more than 3 ga layers.

| ga layers | $AP^{box}$ | $AP^{mask}$ |
|---|---|---|
| $19_{s3}, 31_{s3}, 43_{s3}$ | 56.0 | **49.9** |
| $\mathbf{23_{s3}, 33_{s3}, 43_{s3}}$ | **56.1** | **49.9** |
| $27_{s3}, 35_{s3}, 43_{s3}$ | 55.8 | 49.6 |
| $31_{s3}, 37_{s3}, 43_{s3}$ | 55.9 | 49.8 |
| $41_{s3}, 42_{s3}, 43_{s3}$ | 55.8 | 49.5 |

(d) **Placement Stride.** We find a stride of 10 going back from the last layer in stage 3 to be the best.

Table 19: **HieraDet Global Attention Layers.** Ablating the placement of global attention layers in Hiera, here for a Hiera-L model. The stage the layer belongs to is denoted by $_{s\#}$. We find the optimal strategy to be placing 3 global attention layers in stage 3 of the model. Note that we use absolute win in these experiments, but use different hyperparameters for training than the main paper.

## D MORE EMBEDDING VISUALIZATIONS

Here we present visualizations of more channels of the position embeddings described in the main paper. To show the generality of this issue, we provide visualization of the publicly released "bugged" Hiera models at different scales: tiny (Fig. 10), base (Fig. 11), and huge (Fig. 12). In each case, we present a random sample of position the position embeddings. Fig. 13 shows more examples of absolute win (as in Fig. 3) for a Hiera-L model we trained with the fix.

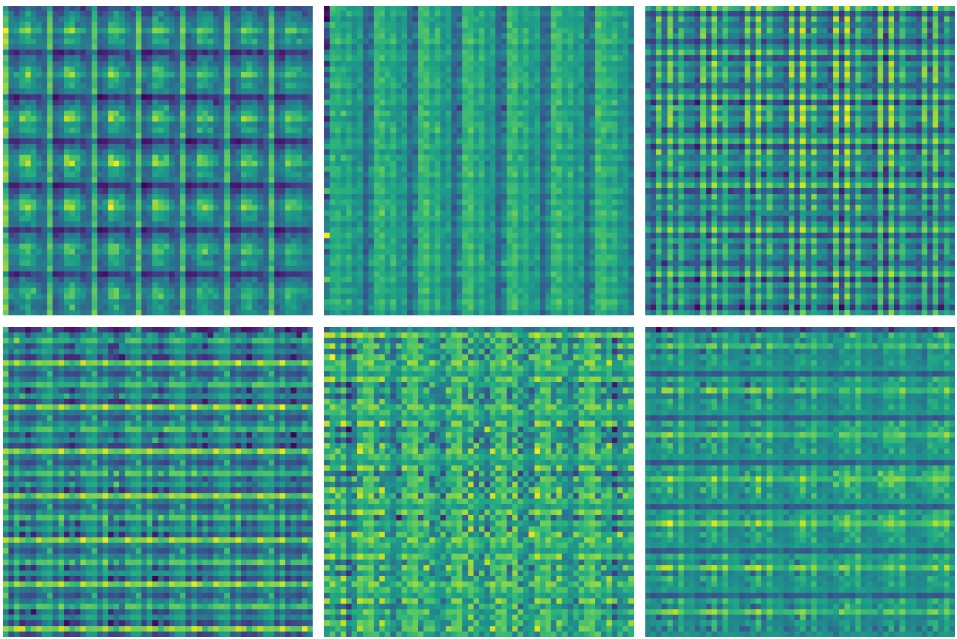

Figure 10: **Hiera-T MAE Image Original Position Embedding.** Channels of the position embedding for the public Hiera-T model pretrained using MAE on images.

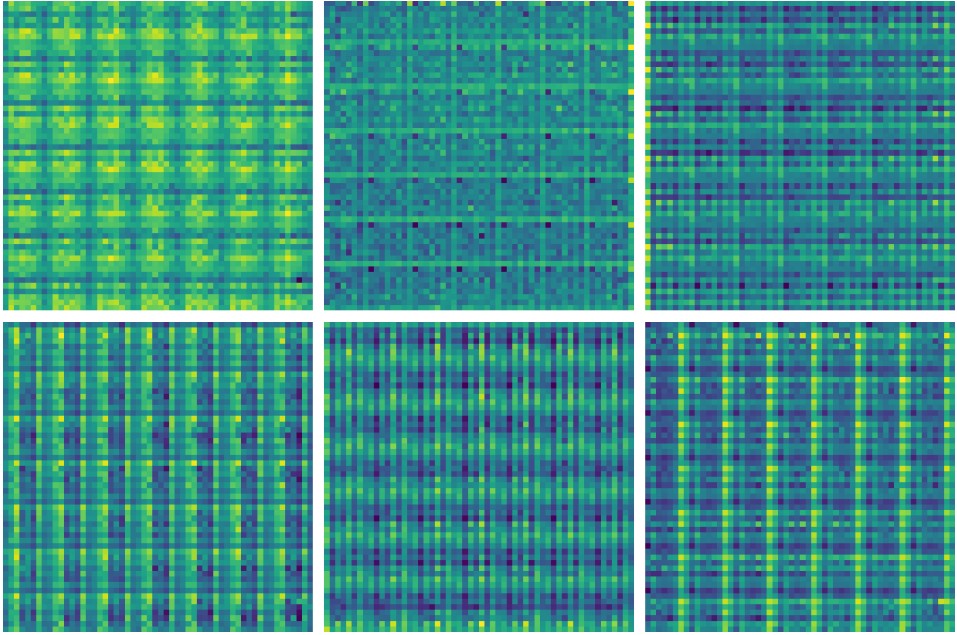

Figure 11: **Hiera-B MAE Image Original Position Embedding.** Channels of the position embedding for the public Hiera-B model pretrained using MAE on images.

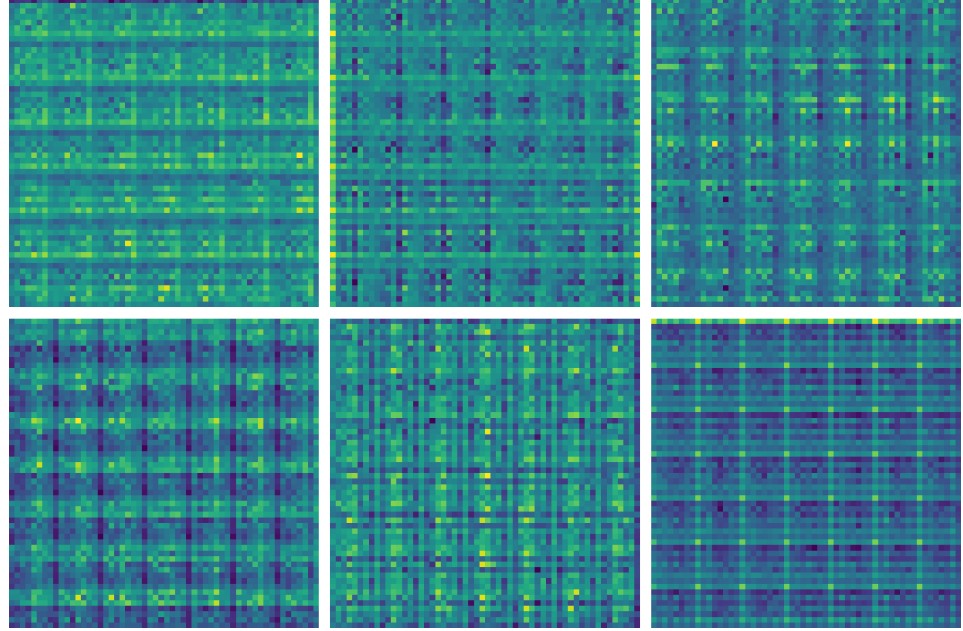

Figure 12: **Hiera-H MAE Image Original Position Embedding.** Channels of the position embedding for the public Hiera-H model pretrained using MAE on images.

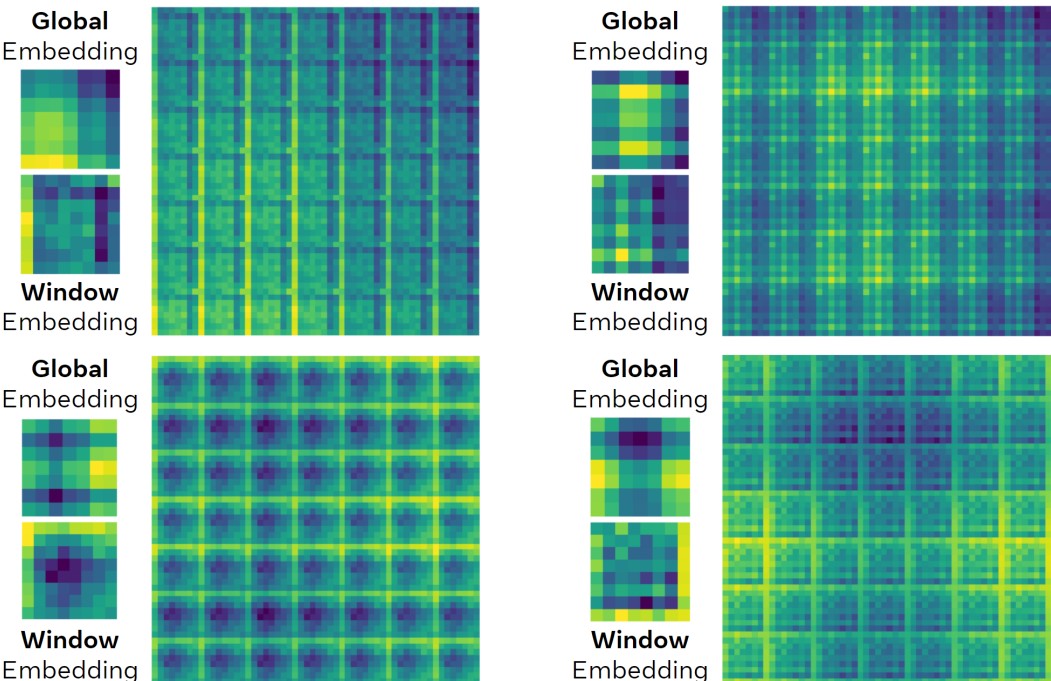

Figure 13: **Hiera MAE Image Absolute Win Position Embedding.** Channels of the position embedding for our Hiera-L model pretrained using MAE on images with absolute win.

## D.1 WINDOW SIMILARITY

While the "windowness" of the original Hiera model's position embeddings are apparent from visualization, we can also measure this issue quantitatively by looking at the average "similarity" between windows at the end of training as in Fig. 4. We've computed this statistic for all the publicly available Hiera ImageNet-1k trained models in Tab. 20:

| Model | T | S | B | B+ | L | H |
|---|---|---|---|---|---|---|
| Window Similarity | 0.83 | 0.81 | 0.84 | 0.77 | 0.74 | 0.66 |

Table 20: **Window Similarity** for the publicly released original Hiera MAE models, measured by taking the average pairwise cosine similarity between all pairs of windows in the position embedding.

While the larger models have less similarity, the similarity is still high for all models. Note that the values in this table are larger than the final number reached in Fig. 4 because we trained those models using 400 MAE epochs, whereas the original Hiera models train for 1600 MAE epochs.

# E   THEORETICAL JUSTIFICATION FOR ABSOLUTE WIN

We've showcased the existence of a bug between interpolating position embeddings and window attention and how absolute win fixes it empirically. In this section, we present additional theoretical justification for why the bug happens and how to fix it.

## E.1   LINKING POSITION EMBEDDINGS TO ATTENTION

In the main paper, we treat the position embedding and the action of attention as intrinsically linked. Here, we show mathematically why this is the case.

Absolute position embeddings (e.g., in Hiera or ViT) are added directly after patchification. For each patchified token at position $i$ with $n$ features, $x_i \in \mathbb{R}^n$, the result of adding the corresponding position embedding, $p_i$, is

$$x'_i = x_i + p_i \tag{1}$$

This $x'_i$ is immediately passed into the first block of the network and thus the first attention layer. For the attention matrix (ignoring heads without loss of generality), the model constructs queries $q_i$ and keys $k_i$ as $q_i = x'_i W_q^T$ and $k_i = x'_i W_k^T$ for weight matrices $W_q, W_k \in \mathbb{R}^{n \times n}$. The (i, j)th element of the attention matrix pre-softmax is then

$$q_i k_j^T = x'_i W_q^T W_k x'^T_j \tag{2}$$

For brevity, let $W_{qk} = W_q^T W_k$. Then, if we substitute back $x_i$, we get:

$$q_i k_j^T = (x_i + p_i) W_{qk} (x_j + p_j)^T \tag{3}$$

When we expand the above expression, we get 4 terms—a term without posemb, 2 cross terms, and the following explicit position embedding term:

$$q_i k_j^T = \ldots + p_i W_{qk} p_j^T \tag{4}$$

Importantly, the same $W_{qk}$ is being used for each position, so the only spatially varying parameters in this term come from the position embeddings. Thus, any repeated pattern we see in the actual values of $p_i$ and $p_j$ (what we visualize in Fig. 2, with similarity plotted in Fig. 5) explicitly affects the action of attention in this first layer of the network. Then, because of skip connections, it also does so in subsequent layers (though with less effect in deeper layers). This is consistent with the commonly held belief that position embeddings have more effect at the start of the network than at the end.

## E.2   HOW REPEATED POSITION EMBEDDINGS CAUSE THE BUG

Following Sec. E.1, we can emulate what happens to cause the bug by explicitly defining $p_i$ to be periodic. For simplicity, we will treat $x$ and $p$ as 1D in space, but this can be extended to any number of dimensions.

Let $\ell$ denote the length of this 1D image used during *pretraining*. Then, let us further assume that the model learns a periodic pattern for $p$ that repeats every $t$ tokens. Then, during finetuning, we upsample the image size to $\ell'$. If we upsample the position embeddings by interpolation, then notably,

the periodicity of the position embeddings will change, i.e. $t' \neq t$. However, the *proportion* each period takes up in the image will remain the same, i.e. $t'/\ell' = t/\ell$.

Now, let us turn our attention to *attention*. Consider a model that uses window attention with a window size of $w$, which doesn't change during finetuning. Assuming the window attention is what caused the periodicity in the position embeddings, then $w = t$ (which is true empirically). Then by Eq. 4, the position bias for the window starting at position $i$ would be

$$
\begin{bmatrix}
p_i W_{qk} p_i^T & p_i W_{qk} p_{i+1}^T & \cdots & p_i W_{qk} p_{i+w}^T \\
p_{i+1} W_{qk} p_i^T & p_{i+1} W_{qk} p_{i+1}^T & \cdots & p_{i+1} W_{qk} p_{i+w-1}^T \\
\vdots & \vdots & \ddots & \vdots \\
p_{i+w-1} W_{qk} p_i^T & p_{i+w-1} W_{qk} p_{i+1}^T & \cdots & p_{i+w-1} W_{qk} p_{i+w-1}^T
\end{bmatrix}
\tag{5}
$$

For brevity, let's consider just one row of this matrix, specifically $p_i W_{qk} \begin{bmatrix} p_i & \cdots & p_{i+w-1} \end{bmatrix}^T$. To reason about this vector, we will make one additional assumption about what the position embeddings learn. And that is, *adjacent* position vectors in a window learn similar values ($p_i \sim p_{i+1}$) and far away positions learn dissimilar values ($p_i \nsim p_{i+w-1}$). Note that the exact specifics of what is exactly similar to what is not explicitly a requirement for the derivation, just this general fact that things inside the window are treated differently to those across the window. But we can see a similar trend empirically in Fig. 5 window similarity. The tokens at the edges are not similar to tokens of the opposite edge, and spatially close tokens are more similar.

Then, observe that in this case our row $p_i W_{qk} \begin{bmatrix} p_i & \cdots & p_{i+w-1} \end{bmatrix}^T$ will have high magnitude values along the first couple of elements, and then lower values for the remaining elements. Now, let's focus on what happens in the higher resolution image, where $w$ remains the same, but $t' = (\ell'/\ell)\, t$.

Presume $t' = t + 1$. Now let's again consider our vector: $p_i W_{qk} \begin{bmatrix} p_i & \cdots & p_{i+w-1} \end{bmatrix}^T$. When $i = 0$, the situation doesn't change much: $p_0$ and $p_1$ are similar, and $p_{w-1}$ is still dissimilar to $p_0$. However, the situation changes when we look at the next window. $p$ is now periodic with a period of $t'$, which is $t + 1 = w + 1$. Thus, $p_w$ is not $p_{t'}$, but rather $p_{t'-1}$. So in the *next* window, $p_w W_{qk} \begin{bmatrix} p_w & \cdots & p_{2w-1} \end{bmatrix}^T$, the two spatially adjacent $p_w$ and $p_{w+1}$ are actually $p_{t'-1}$ and $p_{t'}$, which, modulo the period of $t'$ is the same as comparing $p_0$ and $p_{t'-1}$. Thus, these two adjacent tokens in the window now have a position bias in attention that says they are *not* specially adjacent, and in fact *on the other side of the window from each other*. Moreover, if we take the two farthest tokens in that window, with position embeddings $p_w$ and $p_{2w-1}$, modulo $t'$ this becomes $w_{t'-1}$ and $p_{t'-3}$ which (if $t' > 3$) are considered *similar*.

This continues with different patters for all other windows, meaning that whatever position bias the model learned to rely on during pretaining is now completely lost for finetuning. It is not surprising then, that the accuracy of the model greatly suffers when this happens. Thus, this mismatch between $t'$ and $w$ *is* the bug.

One interesting property of this result is that, if $t'$ is a multiple of $w$, the bug has less of an effect, as the attention bias matrices will then repeat over the windows with a period of $t'/w$, rather than shift in every window. Conversely, the effect of the bug is the greatest when $t'$ is *slightly* different than $w$, as shown in the above example. We can see this empirically in Tab. 8, where 256px images (pretrained on 224px) show the largest drop in performance for the bugged model.

### E.3    HOW TO FIX THE ISSUE

The bug is caused by interpolating the position embeddings, which results in $t' \neq t$. The simplest way to alleviate the issue, then, is just to ensure that $t' = t$. We do this in two ways for absolute win: first, window position embeddings explicitly set $t' = t$, as we manually define it to be periodic with a period of $t = w$, and this doesn't change when the image size changes. Second, our global position embedding can explicitly forbid the embeddings from learning a periodic pattern in the first place by being too small to be affected by the tokens inside of a window. In this case, the model does not have the stipulation that $p_i \nsim p_{i+w-1}$, and instead changes gradually such that the shifting of windows do not invert what was learned during training. As these are orthogonal solutions, we add them to create absolute win.

We choose these solutions because they are simple and they closely resemble what the original model learns, meaning it can be used as a drop-in replacement. However, there are other potential solutions we don't consider in the paper. Specifically, from the derivation in Sec. E.2 hints at another solution that can still be periodic. Instead of having the model learn $p_i \not\propto p_{i+w-1}$ (i.e., periodic with harsh boundaries), we could impose some sort of regularization to ensure that the periodicity is *continuous*. That is, $p_i p_{i+w-1}^T \approx p_i p_{i+1}^T$. If we enforce the model to learn a smooth transition, the attention bias will change less from pretraining to finetuning. However, this type of approach would likely require choosing a good loss function and tuning hyperparameters to get the same accuracy as the original model. Thus, we leave this direction for future work to explore.

Alternatively, we could simply increase the window size in the larger resolution model, i.e. set $w' = (\ell'/\ell) w$. This would also work, but creates a more expensive model (as attention is now over more tokens), and is not actually always possible. Since $w$ has to be an integer (we can't reasonably attend over half a token), this only works for specific resolutions.

## F    PRETRAINING AT HIGHER RESOLUTIONS

In the main paper, we explore increasing the input resolution of Hiera models by pretraining with MAE using the standard 224px image resolution and finetuning at different downstream resolutions. Would it be beneficial to pretrain at the same resolution we finetune at? In this section, we perform some preliminary experiments to answer that question, both for finetuning on ImageNet-1k and for COCO detection.

We first begin by pretraining and finetuning at the same resolution on ImageNet-1k. In each case, we use the same window size (8) and adjust the background position embedding size accordingly (i.e., 14 for 224, 16 for 256, and 24 for 384). We present these results in Tab. 21.

| model | pretrain mask% | pretrain res | finetune res | in-1k acc |
|---|---|---|---|---|
| Hiera-L | 60 | 224 | 256 | **86.5** |
| | 60 | 256 | 256 | 86.4 |
| Hiera-L | 60 | 224 | 384 | 86.9 |
| | 60 | 384 | 384 | 86.8 |
| | 70 | 384 | 384 | **87.0** |
| | 80 | 384 | 384 | 86.9 |
| | 90 | 384 | 384 | 86.4 |

Table 21: **High-Res Pretraining on ImageNet-1k** using Heira-L with absolute win. We also vary the pretrain masking ratio in order to increase task difficulty.

However, the result is that pretraining at a higher resolution actually *hurts* performance, especially if you don't change the task difficulty of the MAE pretraining step. We can get some minor gains for 384px images by pretraining and finetuning with 384 and using a 70% masking ratio, but this is not nearly worth it for the extra pretraining time.

This result isn't too surprising. Prior work (Xie et al., 2022) has found that pretraining at a slightly lower resolution than finetuning performs better on ImageNet-1k using other architectures. But while pretraining at a higher resolution may not be beneficial for ImageNet-1k, it could have a much greater effect on downstream tasks like detection, where the resolutions are much higher than anything used in image classification.

Thus, we also finetune these higher resolution models on COCO and report results in Tab. 22. But note that there's an additional axis of consideration here: when applying absolute win for detection, we tile the original position embedding and replace the global attention with window attention to match that tiling. So what resolution do we construct the original position embedding with? Using a bigger resolution means bigger tiles and more pairwise computation in each window.

And here, the trend reverses: the models pretrained with higher resolutions that performed worse on ImageNet-1k perform a noticable amount better on COCO, even *with the same tiling resolution*. This is important because despite the higher cost of pretraining, models with the same tiling resolution cost the same amount to train and evaluate on detection. So in the case of pretraining with 256 res

| model | pretrain mask% | res | finetune tile res | Mask R-CNN $AP^{box}$ | $AP^{mask}$ |
|---|---|---|---|---|---|
| Hiera-L | 60 | 224 | 224 | 56.2 | 50.0 |
| | | 256 | 224 | **56.6** | **50.1** |
| | | 384 | 224 | 56.4 | 50.0 |
| Hiera-L | 60 | 224 | 256 | 56.3 | 49.8 |
| | | 256 | 256 | **56.5** | **49.9** |
| Hiera-L | 60 | 244 | 384 | 56.2 | 49.8 |
| | | 384 | 384 | **56.5** | **50.0** |

Table 22: **High-Res Pretraining for COCO** using the Heira-L pretrained on ImageNet-1k with absolute win in Tab. 21. Tile res denotes the resolution the position embedding is tiled at as well as the window size for window attention (which are equivalent when using absolute win). Note that a higher tile resolution means a slower model, so good performance at low tiling resolutions is ideal.

and finetuning on detection with a tiling res of 224, we essentially got **+0.4 box** and **+0.1 mask** AP for free at finetune / inference time!

It's likely possible to use this technique to outperform our results in Tab. 11. And moreover, this approach would likely work for other backbones, e.g. with ViTDet, as well. Simply pretrain at 256, downsample the position embedding to 224, and then finetune on detection with a position embedding and window tiling of 224. We leave this for future work to explore.

