# OpenReview forum: "Window Attention is Bugged: How not to Interpolate Position Embeddings"
_ICLR.cc/2024/Conference — ICLR 2024 poster_

### Official Review · Reviewer_4xm4 · 2023-10-29

**Soundness:** 3 good
**Presentation:** 3 good
**Contribution:** 2 fair
**Rating:** 6
**Confidence:** 3

**Summary:**

This paper studies a new problem that there could be a misalignment in the learned position embedding when fine-tuning a model at a higher resolution, the model is composed of both window-attention and absolute position embedding. The solution is to applied separate embeddings for window and global position. The proposed separate position embedding can improve the suffered models, Hiera and ViTDet for classification and object detection respectively.

**Strengths:**

Position embedding is widely used in modern Transformer-based vision architectures. This paper aims at studying important questions on position embedding, and attention-mechanism. The draft is well-written and organized, the proposed method is clearly demonstrated.

**Weaknesses:**

The paper is well-presented, but some issues could be further discussed or studied to avoid misleading. A list of questions is shown below, a short summary for the weaknesses is:  1. The whole idea is based on the visualization of the position embedding, but if the visualization can represent position information is a question, I hope the draft can further explain this question to make the draft clear. 2. The proposed method utilizes the "bug" of the learned window position embedding, which raises a question of if the learned embedding is a bug.
This draft could be summarized as HieraV2 which uses both global and window position embedding based on the feature in Hiera.

**Questions:**

There exist weaknesses in the current draft, which should be discussed further before publication.

1. The motivation:
The whole problem(bug) is found and discussed based on the "learned" feature embedding, which is shown in Figs. 1 and 2 in the draft. And this example plays a foundation role in this study, why it is a bug and what the fix is. However, I am trying to figure out why the visualization can be considered position information? This study uses Hiera as an example to showcase the bug, and Hiera is a modified version of MViTv2. I know some of the learned position embedding also goes through or fused with semantic features, so how can one confirm the defined position embedding is really position embedding? Could the learned block also contain semantic feature and could the semantic feature be the reason behind?(Please correct me if I am wrong on Hiera, MViTv2 uses decomposed position). Assuming if it is position embedding, how one confirm the bug is from window-attention + absolute position? Could the learned embedding is the main reason, in another word, what if I use pre-defined absolute position information with window-attention, could the bug still be there?(I guess the performance could drop, but the claim was on exposing the bug.)

2. The fix:
This study proposes to use separate position embedding for window and global for the bug. This solution could be trivial or the demonstration of the question can be further studied. Hiera is a simplified version of the model, MViTv2, and Hiera replaced relative position with absolute position on purpose because they did not observe differences in their ablation table. Empirically, if mask unit(window-att)+ absolute position could lead to lower performance on higher resolution images, one can simply use relative position as used in original MViTv2. This proposed method can be formulated as combining both window and global position for a slightly higher performance, as shown in Tab 2 in the draft. A further question is that if the method is proposed based on the "bug" not fixing it, is it still a "bug"?

---

> ### Author Response · Authors · 2023-11-18
> **Author Response to Reviewer 4xm4 (1/2)**
>
> We thank the reviewer for their review. We would like to note that the other reviewers agree that we “clearly analyze an important bug”, provide a “comprehensive analysis of the problem” and “propose a simple and effective solution”. Thus we believe there may be some misunderstandings here that we’d like to address. We hope to satisfy the discussion points laid out by the reviewer with the following response.
>
> **Clarification: What the bug is.**
> We believe there might be some confusion as to what the bug actually is. We do not claim that the _windowed position embeddings_ (of e.g., Hiera) is the bug. Instead, the bug is _not accounting for these windows_ when interpolating the position embedding. This causes a mismatch between the windows of the position embeddings and the windows in attention that results in poor performance. It is this _mismatch_ that’s the bug (see Fig. 1), and we fix it by carefully accounting for the windows in each case. We have further elaborated on this in Appendix E.2 in the updated draft of the paper.
>
>
> **W1. The Motivation.**
> These are good points, but we would like to emphasize that the link between position embeddings and attention is broadly treated as common knowledge in the transformer literature (we provide a derivation of this in A below), and as mentioned above, the bug is not that the position embeddings are tiled, but that _not accounting for that tiling when interpolating_ causes an issue (see B below). We elaborate on both points below.
>
> **_A.) It’s not clear that the position embeddings we visualize and their actual effect on the network after passing through multiple layers are the same._**
> They aren’t exactly the same, but they _are_ inherently linked. Note that analyzing the values of position embedding to infer its effect on the rest of the network is a common practice in the ViT literature (e.g., [1,2]). Thus, proving this connection is out of the scope of this work, but we think this is an important discussion, so we’ve described our perspective here.
>
> The amount of processing applied to the features after adding the position embedding is minimal in the early layers. Mathematically, the implementation of Hiera / ViT proceeds like this for each token $x_i$ after the patchification step with position encoding for that token $p_i$: $x'_i = x_i + p_i$
>
> This $x'_i$ gets immediately passed into the first block of the network and thus the first attention layer. For the attention matrix (ignoring heads, as they don’t affect the result), it constructs $q_i$ and $k_i$ as $q_i = x'_i W_q^T$ and $k_i = x'_i W_k^T$. The (i, j)th element of the attention matrix pre-softmax is then
>
> $$q_i k_j^T=x'_i W_q^T W_k {x'}_j^T$$
>
> For brevity, let $W_{qk} = W_q^T W_k$. Then, if we substitute back $x_i$, we get:
>
> $$q_i k_j^T = \left(x_i + p_i\right)W_{qk} \left(x_j + p_j\right)^T$$
>
> When we expand the above expression, we get 4 terms---a term without posemb, 2 cross terms, and the following explicit position embedding term:
>
> $$ q_i k_j^T = … + p_i W_{qk} p_j^T $$
>
> Note that the same $W_{qk}$ is being used for each position here, so the position embeddings are the only parts of this term that vary spatially. Thus, any repeated pattern we see in the actual values of $p_i$ and $p_j$ (what we visualizing in Fig. 2, with similarity plotted in Fig. 5) is very closely reflected in this first layer of the network, and because of skip connections, also in subsequent layers. This is consistent with the commonly held belief that position embeddings have more effect at the start of the network than at the end.
>
> Again, we view this to be out of scope for this work (as this kind of analysis is common), but we’ve added this discussion to the appendix as Appendix E.1 to further solidify our claims. We also have a similar derivation for the bug in Appendix E.2 and an explanation of how absolute win fixes the bug in Appendix E.3. Please take a look if you’re interested.

---

> > ### Comment · Reviewer_4xm4 · 2023-11-21
> > **response**
> >
> > Thanks a lot for the explanation of the background, I can see this is the basic transformer mechanism, which falls into the question raised. The gradient for the learnable POSITION embedding pi is through xi_prime, which means it is not clear how much POSITION or semantic information it actually holds. I can give that the learned feature(hopefully it is position) embedding pi is tiled and shifted after interpolation, and this could be a problem and this study is trying to showcase this problem.
> >
> > The answers are reasonable to me, the solution is a little trivial, bringing back the global attention as an addition to the window one. I would change my rating to 6.

---

> ### Author Response · Authors · 2023-11-18
> **Author Response to Reviewer 4xm4 (2/2)**
>
> **_B.) Assuming position embeddings are the source of the bug, it’s not clear that this is because of window attention—it could be learned position embeddings instead._**
> The reviewer provides a potential counterexample experiment, e.g., if we train a Hiera model with a fixed absolute position embedding (e.g., sinusoidal), would we still have this bug? The answer is (likely), no. In fact, we already have a similar ablation in Tab. 4 (“global only”). This is because the bug is specifically in the mismatched window sizes between these learned position embeddings and the actual windows used when running the model on bigger images. If the position embedding is explicitly not windowed, the model will probably learn a way around this (but with likely lower accuracy, as the reviewer mentioned).
>
> But, that doesn’t mean you can just ignore the bug and use fixed embeddings. Specifically, the ViTDet models actually do use fixed sinusoidal embeddings (from the pretrained MAE models). And yet they still suffer from the bug, not because the position embedding is windowed, but because they specifically add windows which now cease to align with the pre-existing position embedding. So in our method, we manually add windows to the position embedding by tiling them to fix it.
>
> This is the core of the bug, which is described in Fig. 1—a mismatch between windows. The primary goal of this work is to alert the community of this bug so that they can avoid it in their own designs. We hope the reviewer can agree that we provide a significant amount of empirical verification that the bug does exist and that our method can correct it in both Hiera and ViTDet.
>
> [1] On the Relationship between Self-Attention and Convolutional Layers. ICLR 2020.
> [2] Vision Transformers provably learn spatial structure. NeurIPS 2022.
>
>
>
> **W2. The fix.**
> Again, we would like to point out that the bug is in a mismatch between the size of windows in the position embedding and the size of window attention. Relative position embeddings can recover the lost accuracy, but this is at the cost of a large speed penalty that absolute win avoids. Furthermore, absolute win leans into the _tiling behavior_ of window attention, not the bug. As explained in Appendix E.3 and shown visually in Fig. 1, it explicitly fixes the bug. We elaborate on these points below.
>
> **_A.) Hiera removed relative position embeddings, adding them back could fix it._**
> This is very true, and is actually what the original ViTDet does. Because of the bug, the original ViTDet requires relpos to get good accuracy. Which is fine for accuracy, but relpos is significantly slower than not having relpos. If we are just careful about how we define position embeddings (abs win in our paper), we can completely avoid having to add relpos and take the speed penalty _and_ get a further accuracy boost. The same would likely be true for Hiera, adding back the relpos might alleviate some of the accuracy drop due to the bug, but it would make the model significantly slower and just like ViTDet, the accuracy would likely still be worse.
>
> Regardless, this is exactly the kind of situation we want to help avoid with this paper. By showing that this bug is a possibility, we hope that the community can avoid future issues like this when designing new models and architectures.
>
> **_B.) If the method is proposed based on the "bug" not fixing it, is it still a "bug"?_**
> We think there may be some confusion here. We _do_ fix the bug, as very clearly shown empirically through our comparisons to the original model (and we further elaborate in the updated Appendix E.3). The bug is as shown in Fig. 1, an induced misalignment between the size of windows in the position embedding and the size of windows in window attention. By explicitly windowing our position embedding (absolute win), we can exactly control these two windows sizes to perfectly align at any image resolution, which means that there’s no mismatch and thus no bug.
>
>
>
>
> We thank the reviewer for inviting this discussion. We hope our response has cleared up some confusion as to what we are defining as the “bug” and for how we motivate the existence of the bug. If there are any lingering questions, we invite the reviewer for further discussion. And once again, we would like to thank the reviewer for their time and feedback.

---

### Official Review · Reviewer_7g6r · 2023-11-01

**Soundness:** 3 good
**Presentation:** 3 good
**Contribution:** 3 good
**Rating:** 6
**Confidence:** 3

**Summary:**

This paper identifies and fixes a bug that occurs when using both window attention and absolute position embeddings in vision transformers. The authors show that naively interpolating absolute position embeddings when finetuning on larger image sizes misaligns the embeddings with the window attention, hurting performance.

The bug is analyzed in detail using Hiera and ViTDet as case studies. Hiera learns repetitive absolute position embeddings aligned with its window attention during pretraining. Interpolating these embeddings when finetuning on larger images misaligns them, causing a drop in accuracy.

ViTDet suffers from a similar issue - it uses interpolated absolute position embeddings with added window attention, which provides each window only a sliced portion of the full embedding.

To fix the bug, the authors propose "absolute win" - embedding strategies that align position embeddings with window attention after interpolation. This involves learning separate window and global position embeddings.

On Hiera, absolute win embeddings fix the bug and achieve strong image classification accuracy when finetuning on larger resolutions. The embeddings also enable removing most relative position embeddings from ViTDet while increasing performance.

The authors then apply absolute win to create HieraDet for detection, outperforming ViTDet by 1.5+ box AP on COCO using only ImageNet pretraining.

**Strengths:**

- Identifies and clearly analyzes an important bug in modern vision transformers
- Proposes a simple and effective solution - "absolute win" position embeddings
- Achieves SOTA detection performance with HieraDet using only ImageNet-1k pretraining
- Improves performance across multiple architectures (Hiera, ViTDet) and tasks

**Weaknesses:**

- The gains from fixing this bug seem somewhat task/architecture dependent (smaller gains on ViTDet).
- Unclear if the bug manifests in other architectures with window attention and absolute embeddings.
- Does not explore more powerful global position embedding strategies for ViTDet.

**Questions:**

Does relative position encoding exhibit similar phenomena as absolute position encoding?

---

> ### Author Response · Authors · 2023-11-18
> **Author Response to Reviewer 7g6r**
>
> We thank the reviewer for their extensive summary and concise enumeration of the paper’s strengths that it “clearly analyzes an important bug”, “proposes a simple and effective solution”, “achieves SOTA detection performance”, and “improves performance across multiple architectures”. We would like to address the reviewer’s concerns below.
>
>
> **W1. Gains seem task/architecture dependent.**
> This is a good observation, and there’s a good reason for it. For ViTDet, the bug actually does also cause a 1 mAP drop (similar to HieraDet), but this drop is _hidden_ behind ViTDet’s choice to add extra relative position embeddings. In the original ViTDet paper [1], the authors make a comment (page 11 on the arxiv version) that this addition improved AP by ~1 point. What we show in this work is that by fixing the bug using a more principled approach, we can get that same 1 mAP back (and even more) without those expensive relative position embeddings (thereby increasing inference speed by ~40%). So, in reality, the bug itself has a similar effect in most cases—it’s just that ViTDet’s original design obscured that effect.
>
> [1] Exploring Plain Vision Transformer Backbones for Object Detection. ECCV 2022.
>
>
> **W2. Unclear if bug manifests in other architectures.**
> This is something we also don’t know for sure. But our goal is not necessarily to expose an issue in _every_ method out there. Instead, using two comprehensive recent examples (Hiera and ViTDet), we want to make the fact that this issue exists abundantly clear to the community so that they can fix it or avoid causing it in their own work—pushing the SotA is just a bonus.
>
>
> **W3. Does not explore more powerful global position embedding strategies for ViTDet.**
> The scope of this work is primarily about the bug and fixing it. More study into whether relpos is best for the global embedding for ViTDet (other than the existing Tab. 4 ablation) would have been interesting, but we didn’t want to stray too far from the original ViTDet model in practice.
>
>
> **Q1. Does relative position encoding exhibit similar phenomena as absolute position encoding?**
> This is a good question and a worthy line of inquiry for future work. Without such an investigation, we can only speculate, but relpos is likely a little more resilient to this issue than absolute embeddings. Fundamentally, relpos recomputes and resamples the position embedding depending on the _relative_ position between two tokens. Since there’s a layer of abstraction between the actual position embeddings and the positions of the tokens themselves, it can be recomputed correctly for different image sizes. However, this abstraction is also what makes relpos slow—so, there’s a tradeoff there.
>
>
> We would like to once again thank the reviewer for their time.

---

### Official Review · Reviewer_QscK · 2023-11-02

**Soundness:** 3 good
**Presentation:** 4 excellent
**Contribution:** 2 fair
**Rating:** 6
**Confidence:** 4

**Summary:**

The authors discuss the challenges faced when combining window attention, position embeddings, and high-resolution finetuning in the modern transformer era of computer vision. They identify a bug that arises when interpolating position embeddings while using window attention. Two state-of-the-art methods, Hiera and ViTDet, are found to be affected by this bug. To address this, the authors introduce an "absolute window position embedding strategy" which rectifies the issue in Hiera and enhances the speed and performance of ViTDet.

**Strengths:**

In-depth Analysis: The paper provides a comprehensive analysis of the problem, identifying the root cause of the issue with window attention and position embeddings.

Solution-Oriented: The authors not only identify the problem but also propose a solution in the form of an "absolute window position embedding strategy".

Empirical Evidence: The paper presents empirical results, showcasing the effectiveness of their proposed solution, especially with the HieraDet model's performance on COCO.

Relevance: The topic is highly relevant given the prominence of transformer architectures in computer vision.

**Weaknesses:**

Limited Scope: The study primarily focuses on two methods, Hiera and ViTDet. Expanding the scope to include more methods might provide a broader understanding of the issue.

Assumption-based: Some conclusions, especially regarding the behavior of position embeddings, seem to be based on observations and assumptions. More rigorous testing might solidify these claims.

Furthermore, the paper's contribution appears to be limited. It primarily addresses a minor bug in Hiera and ViTDET. The resulting improvements, while noteworthy, are not substantial, suggesting that the issue wasn't fundamental. Nonetheless, I appreciate the effort and contribution of this work

**Questions:**

Please see above.

---

> ### Author Response · Authors · 2023-11-18
> **Author Response to Reviewer QscK**
>
> We thank the reviewer for their thoughtful review, and for their kind comments that “the paper provides a comprehensive analysis of the problem”, “propose a solution”, and “showcase the effectiveness” for a “highly relevant” topic. We understand the concerns the reviewer has brought forth, and would like to address them below.
>
> **W1. Limited scope (Hiera and ViTDet).**
> We definitely agree that including more methods could lead to a better understanding of the issue. However, we want to emphasize that our primary goal is to ensure the broader community knows about this potential issue when designing their own approaches, and therefore we have opted to do a _deep, comprehensive_ analysis of fewer, but recent state-of-the-art methods over a shallower, broader study. Readers may find it surprising that the same bug that affects Hiera (a classification model) also affects ViTDet (a detection approach), with each methods’ problems manifesting in different ways. We believe this is broad enough for the reader to get an understanding of the issue and for them to be able to avoid it in their own work. Certainly, looking at more methods could help, but as the reviewer has stated, we already provide “a comprehensive analysis of the problem” with just these two methods. So, we hope that this is not a make or break weakness of the paper.
>
> **W2. Assumption-based (Behavior of position embeddings based on observation / assumptions).**
> It is very true that we take a more empirical / observation-based approach in this paper. This is because the issue with the position embeddings (especially for Hiera), is _extremely_ apparent when visualizing them. In the appendix (Fig. 10) we provide more examples of the original position embeddings to show that behavior. And, if one doesn’t believe us, it’s easy to download an existing Hiera model and plot the position embeddings to get the same result. It really is every Hiera model across _every_ model size that exhibits this behavior, which is why our solution can be so targeted to fixing the exact problem.
>
> However, we do agree that this ubiquity is not fully explored in the initial draft (mostly due to space constraints). To address this in the updated draft, we’ve expanded Appendix D to include visualizations of the position embeddings for more models (Fig. 10-12), as well as the following table in Appendix D.1 that shows the “window similarity” (as described Fig. 4 and Appendix B) for all Hiera image models from the official Hiera release:
>
> | Model             |   T  |   S  |   B  |  B+  |   L  |   H  |
> |-------------------|:----:|:----:|:----:|:----:|:----:|:----:|
> | Window Similarity | 0.83 | 0.81 | 0.84 | 0.77 | 0.74 | 0.66 |
>
> While larger models have lower similarity, even Hiera-H has highly similar windows. We hope this update reinforces that this trend is not just an assumption but instead is a wide-spread issue.
>
> Moreover, we’ve also added a theoretical derivation of what the problem is and how absolute win fixes it in Appendix E of the updated draft in order to strengthen the motivation for our approach. Please take a look if you are interested.
>
> **W3. Limited contribution / non-substantial improvements.**
> While this may be a minor bug, the resulting performance differences are anything but minor. For instance HieraDet-B with Mask-RCNN (Table 10) goes from 52.2 box mAP to 53.7 box mAP, a +1.5 mAP jump on a dataset where the difference in performance between top methods are between 0.3 and 0.9 mAP (see COCO’s leaderboard for example). For ViTDet, the improvement is concentrated less in the accuracy (AP) itself, but instead a 43% speed-up (e.g., for L) that comes essentially for free (performance increases and the model becomes simpler, not more complex). We believe the community will find these aspects of this approach very useful, especially in detection where both speed and accuracy matter. Moreover, the broader community would benefit greatly from knowledge of this bug in the first place, regardless of our method, so that they can avoid making the same mistakes in the future.
>
>
> We would again like to thank the reviewer. We do believe their weaknesses are valid and have updated the paper to address them. We hope the reviewer would agree that these weaknesses are not significant enough to impact the paper’s worthiness of publication.

---

> > ### Comment · Reviewer_QscK · 2023-11-22
> > **Thank you for the rebuttal**
> >
> > Dear authors,
> >
> > Thank you for your comprehensive responses, which effectively addressed my concerns. I will raise my score.

---

### Official Review · Reviewer_hUZp · 2023-11-06

**Soundness:** 3 good
**Presentation:** 3 good
**Contribution:** 3 good
**Rating:** 6
**Confidence:** 2

**Summary:**

The paper identifies an issue that arises when using absolute position embeddings with windowed attention in vision transformers. The key points are:

- Many vision transformers use both absolute position embeddings and windowed attention for efficiency. However, naively interpolating the position embeddings when finetuning on larger image sizes misaligns them with the window partitions.

- This causes significant performance degradation when finetuning on larger images, which the paper demonstrates on Hiera and ViTDet models.

- To fix this, the authors propose "absolute win" embeddings, which separately learn an interpolated global position embedding and a tiled window position embedding. This aligns the embeddings with window attention properly.

- Applying absolute win embeddings improves performance substantially in Hiera and ViTDet across tasks like image classification, object detection, video action recognition.

- With the fix, Hiera and ViTDet establish new state-of-the-art results on COCO and other datasets using only ImageNet-1K pretraining, while also being faster due to reduced need for relative position biases.

- The paper provides practical guidelines for using absolute position embeddings with windowed attention when finetuning vision transformers at higher resolutions.

The key contribution is identifying and fixing a bug that improves SOTA vision transformers. The fix is simple but impactful - just changing the position embeddings.

**Strengths:**

The authors identify an important yet overlooked interaction between commonly used components in vision transformers - window attention and absolute position embeddings. When naively combined and interpolated for high-resolution finetuning, they show this leads to degraded performance.

The key novelty is in formally characterizing and providing strong empirical evidence for this bug. While simple in retrospect, it is a creative insight to recognize that the position embeddings effectively "tile" based on the window size due to the weight sharing in attention. The visualizations and analysis of the learned embeddings clearly demonstrate this phenomenon.

Fixing the bug is also quite elegant - simply embrace the tiling behavior with "absolute win" position embeddings. This simple change allows models to properly interpolate and achieves significantly improved performance across various vision tasks.

The paper is very clearly written and easy to follow. The introduction motivates the problem well, and the methodology is laid out in a structured manner. Experiments comprehensively ablate the impact of the bug fix on state-of-the-art methods.

Overall, this is a significant finding - identifying and correcting an important issue affecting a large class of vision transformer models. The clarity and thoroughness of the empirical validation make a compelling case for the prevalence and implications of this bug.

**Weaknesses:**

* The proposed fix of "absolute win" embeddings is intuitive, but the paper does not provide much analysis into why tiling the window embeddings specifically works better than other potential solutions. Some ablation studies on the design choices could add more clarity.
* The simplicity of the proposed fix somewhat limits the depth of technical novelty and contribution. Providing more design motivation and analysis for "absolute win" would strengthen this aspect.
* Tiling the window position embeddings makes sense based on the observed tiling behavior, but other arrangements could also resolve the misalignment issue when interpolating. The design space is not explored in depth.
* While shown to work empirically, the approach lacks theoretical analysis into why tiling embeddings avoids the misalignment issue during interpolation. A more formal understanding could improve the method.

**Questions:**

see weakness

---

> ### Author Response · Authors · 2023-11-18
> **Author Response to Reviewer hUZp**
>
> We thank the reviewer for their comprehensive summary and for their kind words, e.g., that we “provide strong empirical evidence for this bug”, “it is a creative insight”, “fixing the bug is also quite elegant”, and that “this is a significant finding.” We also thank the reviewer for their constructive criticism. We’ve made changes to the draft based on the suggestions and have collected our responses below.
>
>
>
> **W4. While shown to work empirically, the approach lacks theoretical analysis into why tiling embeddings avoids the misalignment issue during interpolation. A more formal understanding could improve the method.**
> Thanks for the suggestion. We’ve added a 2 page Appendix E to the updated draft of the paper, which attempts to more rigorously derive position embedding’s effect on attention, how a mismatch between periodicity in the position embedding and window attention causes this misalignment issue, and how absolute win fixes it. Along with this derivation come findings that the bug is most detrimental when the finetuning resolution is slightly different than the pretraining one, which explains the empirical behavior of the original model, like in Tab. 8. Please take a look at the updated draft if interested.
>
> **W3. Tiling the window position embeddings makes sense based on the observed tiling behavior, but other arrangements could also resolve the misalignment issue when interpolating. The design space is not explored in depth.**
> The goal of this work is to identify this issue and propose a fix so that the broader community can avoid the bug when designing their own models. To that end, we chose the simplest option that deviates the least from the original design of the model. We agree that there likely are other designs that could also fix the bug. However, we believe this kind of analysis is better saved for follow-up work. We have included additional discussion about this in Appendix E.3.
>
> **W2. The simplicity of the proposed fix somewhat limits the depth of technical novelty and contribution. Providing more design motivation and analysis for "absolute win" would strengthen this aspect.**
> The design for absolute win was simple to attempt to match the existing position embeddings as much as possible while still fixing the bug—specifically so that the fix could be used as a simple drop in replacement. A more complicated approach would have impeded this goal. As for motivation and analysis, we have provided theoretical justifications in Appendix E.2 and E.3 of the updated draft.
>
> **W1. The proposed fix of "absolute win" embeddings is intuitive, but the paper does not provide much analysis into why tiling the window embeddings specifically works better than other potential solutions. Some ablation studies on the design choices could add more clarity.**
> In the updated draft, we’ve backed up this intuition with mathematical justification (see Appendix E), which should hopefully show why this is a natural choice to fix the bug. We also have ablations in Tab. 2 that show two different methods to fixing the bug: tiling (window only) and using a position embedding small enough such that all tokens in a window map to the same location in the embedding (global only). Both work to fix the issue, but because Hiera has both window attention layers and global attention layers, we need to add both embeddings to retain the accuracy of the original model at the original resolution.
>
>
> We would like to again thank the reviewer for their time for helping strengthen the paper with their suggestions.

---

### Meta-Review · Area_Chair_1GQg · 2023-12-05

**Metareview:**

This paper identifies an issue in the interaction of window attention and position embeddings in vision transformers, pinpointing a critical bug when combining these elements. To fix this issue, an absolute window position embedding strategy is proposed. The empirical evidence convincingly demonstrates the existence of the bug and the efficacy of the proposed solution on Hiera and ViTDet.

The reviewers agree that this paper is generally well written, the proposed method is simple and effective, and the empirical study is convincing. The authors also address the reviewers’ concerns in the discussion phase. All the reviewers give positive ratings to this paper. Therefore, we recommend acceptance.

**Justification For Why Not Higher Score:**

While the paper conducts solid empirical study on Hiera and ViTDet, the reviewers point out that the scope of the paper seems relatively limited, and it is not clear whether the conclusion generalizes to more architectures.

**Justification For Why Not Lower Score:**

This paper presents an overlooked issue in Vision Transformers and an effective fix for it. The proposed method achieves strong experimental results. These findings can be interesting to the community.

---

### Decision · Program_Chairs · 2024-01-16

Accept (poster)